# In Defense of Softmax Parametrization for Calibrated and Consistent Learning to Defer

Yuzhou Cao[1]    Hussein Mozannar[2]    Lei Feng[1]*
Hongxin Wei[3]    Bo An[1]

[1]School of Computer Science and Engineering, Nanyang Technological University, Singapore
[2]CSAIL and IDSS, Massachusetts Institute of Technology, Cambridge, MA
[3]Department of Statistics and Data Science, Southern University of Science and Technology, China
`yuzhou002@e.ntu.edu.sg, mozannar@mit.edu`
`lfengqaq@gmail.com, weihx@sustech.edu.cn, boan@ntu.edu.sg`

## Abstract

Enabling machine learning classifiers to defer their decision to a downstream expert when the expert is more accurate will ensure improved safety and performance. This objective can be achieved with the learning-to-defer framework which aims to jointly learn how to classify and how to defer to the expert. In recent studies, it has been theoretically shown that popular estimators for learning to defer parameterized with softmax provide unbounded estimates for the likelihood of deferring which makes them uncalibrated. However, it remains unknown whether this is due to the widely used softmax parameterization and if we can find a softmax-based estimator that is both statistically consistent and possesses a valid probability estimator. In this work, we first show that the cause of the miscalibrated and unbounded estimator in prior literature is due to the symmetric nature of the surrogate losses used and not due to softmax. We then propose a novel statistically consistent asymmetric softmax-based surrogate loss that can produce valid estimates without the issue of unboundedness. We further analyze the non-asymptotic properties of our method and empirically validate its performance and calibration on benchmark datasets.

## 1 Introduction

As machine learning models get deployed in risk-critical tasks such as autonomous driving [18], content moderation [27], and medical diagnosis [20], we have a higher urgency to prevent incorrect predictions. To enable a safer and more accurate system, one solution is to allow the model to defer to a downstream human expert when necessary. The **L**earning to **D**efer (L2D) paradigm achieves this aim [28, 30, 31, 9, 44, 32, 33, 45, 3, 35, 15, 42] and enables models to defer to an expert, i.e., abstain from giving a prediction and request a downstream expert for an answer when needed. L2D aims to train an augmented classifier that can choose to defer to an expert when the expert is more accurate or make a prediction without the expert. L2D can be formulated as a risk-minimization problem that minimizes the 0-1-deferral risk [30], which incurs a cost of one when the classifier is incorrect or when we defer to an expert who errs and incurs a cost of zero otherwise.

Despite being formulated straightforwardly, the risk minimization problem is NP-hard even in simple settings [15, 32] due to the discontinuous and non-convex nature of the 0-1-deferral loss. To make the optimization problem tractable, many efforts have been made to design a continuous surrogate loss for the 0-1-deferral loss while guaranteeing statistical consistency, which means that the minimizer of the surrogate risk is that of the 0-1-deferral risk. In Mozannar and Sontag [30], a cross-entropy-like

---

*Corresponding author: Lei Feng.

37th Conference on Neural Information Processing Systems (NeurIPS 2023).

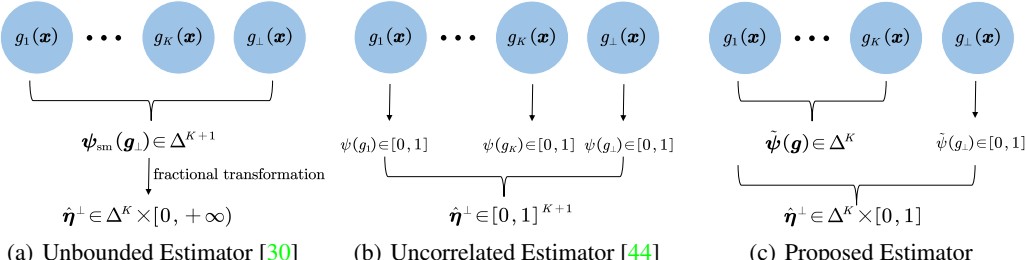

(a) Unbounded Estimator [30]    (b) Uncorrelated Estimator [44]    (c) Proposed Estimator

**Figure 1:** Illustration of the proposed and previous estimators. Probability estimation in L2D aims to predict both the class probabilities and the expert's accuracy $\boldsymbol{\eta} \times \Pr(M = Y|X = \boldsymbol{x}) \in \Delta^K \times [0,1]$, while only our proposed estimator takes exactly the same range.

surrogate loss is proposed that is statistically consistent. Charusaie et al. [9] further generalized the results in Mozannar and Sontag [30] and showed that all the losses that are consistent for ordinary multi-class classification, i.e., classification-calibrated [38, 43, 6], are consistent for L2D with certain reformulation, which makes L2D a more comprehensive framework. By minimizing the risk *w.r.t.* consistent surrogate losses [30, 9], we can obtain a model that defers to who between the classifier and expert is more accurate on an instance basis. However, we also need assessments of the uncertainty of the classifier when it predicts and the uncertainty of the expert prediction when we defer. This allows us to perform better triaging of samples and provide reliable estimates of uncertainty. Verma and Nalisnick [44] pointed out that the previous estimator in [30] generates highly biased probability estimates for the predictions due to its unboundedness. They then proposed a surrogate based on the One-versus-All (OvA) strategy [52] to alleviate this problem, which allows both statistical consistency and improved calibration compared to the previous softmax-based method [30].

Does the difference in performance between OvA and softmax-based methods indicate that softmax parameterization is a sub-optimal choice in L2D? The cause of the unbounded probability estimation in [30] is in fact still elusive, and we cannot simply attribute it to softmax parameterization. Furthermore, notice that the softmax-based method [30] is only a specific implementation of applying softmax in L2D. Meanwhile, softmax parameterization is also a more straightforward modeling of multiclass posterior distributions: while the OvA strategy works by splitting a $K$-class classification problem into $K$ independent binary classification problems and thus induces $K$ uncorrelated probability estimators for each class, softmax parameterization works by directly estimating class posterior probabilities as a whole. Given the wide use and practical advantages of softmax parameterization in classification tasks and the resemblance between L2D and multiclass classification, it is also promising that the usage of the softmax function can result in competitive methods for L2D. Then a natural question arises: can we design a softmax-based consistent surrogate loss for L2D without triggering unbounded probability estimation enabling calibrated estimates?

In this paper, we give a positive answer to this question by providing a novel consistent surrogate loss with *asymmetric softmax parameterization*, which can be seen as the combination of ordinary softmax function and an additional probability estimator. This surrogate can also induce a non-trivial bounded probability estimator for prediction probabilities and expert accuracy. To defend the use of softmax parameterization, we show that it is not the cause of the failure but rather it is the symmetric structure of the surrogates that leads to the unboundedness in probability estimation. Unlike the unbounded estimator [30] that mixes up class probabilities and the expert's accuracy and the OvA estimator that treats each class and expert independently, our method models the class probabilities as a whole with softmax and the expert's accuracy independently with a normalized function, which better reflects the structure of L2D with probability estimation. The differences between our work and previous works are illustrated in Figure 1. We further analyze the limitation and broader impact of our work in Appendix G. Our contributions are four-fold and summarized below:

- In Section 3, we prove that a non-trivial bounded probability estimator does not exist as long as we are using *symmetric* loss functions.

- We propose an asymmetric formulation of softmax parameterization, and show that it can induce both a consistent surrogate loss and a non-trivial bounded probability estimator. We further show that the proposed asymmetric softmax-based method and OvA-based method [44] actually benefit from their asymmetric structure.

- We further study the regret transfer bounds of our proposed asymmetric surrogate and show that it is compatible with both L2D and multi-class classification.

- Experiments on datasets with both synthetic experts and real-world experts are conducted to demonstrate the usefulness of our method for both prediction and expert accuracy estimation.

## 2 Preliminaries

In this section, we review the problem setting of L2D and briefly introduce previous works.

### 2.1 Problem Setting

In this paper, we study the problem of learning to defer to an expert in the $K$-class classification scenario. Let us denote by $\mathcal{X}$ and $\mathcal{Y} = \{1, \cdots, K\}$ the feature and label space respectively. Denote by $X \times Y \times M \in \mathcal{X} \times \mathcal{Y} \times \mathcal{Y}$ the data-label-expert random variable triplet and $\boldsymbol{x}, y, m$ are their realizations, which obeys an underlying distribution with density $p(\boldsymbol{x}, y, m)$. We have access to data triplets $\{(\boldsymbol{x}_i, y_i, m_i)\}_{i=1}^n$ that are drawn independently and identically from the distribution. The goal of L2D in the classification scenario is to obtain a classifier with deferral option $f(\cdot) : \mathcal{X} \to \mathcal{Y}^\perp$, where $\perp$ is the option of deferral to the expert and $\mathcal{Y}^\perp$ is the augmented decision space $\mathcal{Y} \cup \{\perp\}$.

The problem evaluation is the following 0-1-deferral loss $\ell_{01}^\perp$ [2], which is a generalized version of the zero-one loss $\ell_{01}(f(\boldsymbol{x}), y) = [\![f(\boldsymbol{x}) \neq y]\!]$, which takes the value of 1 if we use the output of the classifier when it predicts an incorrect label ($[\![f(\boldsymbol{x}) \in \mathcal{Y} \text{ and } f(\boldsymbol{x}) \neq y]\!]$) or defer to the expert when they are incorrect ($[\![f(\boldsymbol{x}) = \perp \text{ and } m \neq y]\!]$), where $[\![\cdot]\!]$ is the Iverson bracket notation suggested by Knuth [22]. Then we aim to minimize the risk *w.r.t.* to this loss:

$$\min_f R_{01}^\perp(f) = \mathbb{E}_{p(\boldsymbol{x}, y, m)} \left[ \ell_{01}^\perp(f(\boldsymbol{x}), y, m) \right]. \tag{1}$$

Denote by $f^*$ the minimizer of the risk $R_{01}^\perp(f)$, i.e., the Bayes optimal solution, and $\boldsymbol{\eta}(\boldsymbol{x}) = \{\Pr(Y = y | X = \boldsymbol{x})\}_{y=1}^K$. Mozannar and Sontag [30] provide a characterization of the form of the Bayes optimal solution as:

**Definition 1.** (Bayes optimality of L2D) A classifier with deferral option $f^* \to \mathcal{Y}^\perp$ is the minimizer of $R_{01}^c(f)$ if and only if it meets the following condition almost surely:

$$f^*(\boldsymbol{x}) = \begin{cases} \perp, & \max_y \eta_y(\boldsymbol{x}) < \Pr(M = Y | X = \boldsymbol{x}), \\ \operatorname{argmax}_y \eta_y(\boldsymbol{x}), & \text{else.} \end{cases}$$

The form of the optimal solution is intuitive: we should defer to the expert if they have a higher probability of being correct than the optimal classifier which follows the most likely label class given the input $\boldsymbol{x}$. This optimal solution can also be seen as the generalized version of Chow's rule in learning to reject [10], where a fixed cost in $[0, 1]$ serves as the known accuracy of an expert.

### 2.2 Consistent Surrogate Losses for L2D

Although we know the form of the Bayes optimal solution, practically the risk minimization problem above faces computational difficulties: the minimization of the discontinuous and non-convex problem (1) is NP-hard [32]. A widely used strategy for tackling this difficulty is to substitute the original loss function which is discontinuous with a continuous surrogate, which has been applied in many areas including ordinary classification [52, 6, 43, 39, 16, 37], multi-label classification [17, 23, 51, 47], AUC optimization [17, 23, 29], cost-sensitive learning [41, 11], top-$K$ classification [25, 48], adversarially robust classification [4, 2, 1], and learning to reject [12, 13, 5, 49, 34, 8, 7]. Denote by $\boldsymbol{g} : \mathcal{X} \to \mathbb{R}^{K+1}$ the learnable scoring function that induces our decision function for L2D $f : \mathcal{X} \to \mathcal{Y}^\perp$ with the following transformation $\varphi : \mathbb{R}^{K+1} \to \mathcal{Y}^\perp$:

$$\varphi(\boldsymbol{g}(\boldsymbol{x})) = \begin{cases} \perp, & g_{K+1}(\boldsymbol{x}) > \max_{y \in \mathcal{Y}} g_y(\boldsymbol{x}) \\ \operatorname{argmax}_{y \in \mathcal{Y}} g_y(\boldsymbol{x}), & \text{else.} \end{cases}$$

---

[2]Though the formulation introduced here is not its most general version, we only study this one and its induced surrogates (4) since previous works on the design of surrogate losses for L2D [30, 9, 44] also concentrated on it.

We consider a continuous surrogate function $\Phi : \mathbb{R}^{K+1} \times \mathcal{Y} \times \mathcal{Y} \to \mathbb{R}^+$ and the surrogate risk below:

$$\min_{\boldsymbol{g}} R_\Phi^\perp(\boldsymbol{g}) = \mathbb{E}_{p(\boldsymbol{x},y,m)} \left[ \Phi(\boldsymbol{g}(\boldsymbol{x}), y, m) \right]. \tag{2}$$

Assuming we find a surrogate function that overcomes the computational optimization challenges, we need to verify the consistency of $\Phi$ *w.r.t.* $\ell_{01}^\perp$, i.e., any minimizer of the surrogate risk also minimizes the original risk: (2):

$$\varphi \circ \boldsymbol{g}^* \in \operatorname{argmin}_f R_{01}^\perp(f), \quad \forall \boldsymbol{g}^* \in \operatorname{argmin}_{\boldsymbol{g}} R_\Phi^\perp(\boldsymbol{g}).$$

The first consistent surrogate [30] *w.r.t.* $\ell_{01}^\perp$ is proposed by modifying the softmax cross-entropy loss:

$$L_{\mathrm{CE}}(\boldsymbol{g}(\boldsymbol{x}), y, m) = -\log \psi_y^{\mathrm{sm}}(\boldsymbol{g}(\boldsymbol{x})) - [\![ m = y ]\!] \log \psi_{K+1}^{\mathrm{sm}}(\boldsymbol{g}(\boldsymbol{x})), \tag{3}$$

where $\boldsymbol{\psi}^{\mathrm{sm}}$ is the softmax function $\psi_y^{\mathrm{sm}}(\boldsymbol{u}) = \exp(u_y) / \sum_{y'=1}^{d} \exp(u_{y'})$, where $d$ is the dimensionality of the input. Inspired by the risk formulation above, Charusaie et al. [9] further generalized the family of consistent surrogate for $\ell_{01}^\perp$ by considering the following consistent surrogate reformulation:

$$L_\phi(\boldsymbol{g}(\boldsymbol{x}), y, m) = \phi(\boldsymbol{g}(\boldsymbol{x}), y) + [\![ m = y ]\!] \phi(\boldsymbol{g}(\boldsymbol{x}), K+1). \tag{4}$$

It is known from Proposition 2 in Charusaie et al. [9] that surrogates take the form above are consistent *w.r.t.* $\ell_{01}^\perp$ if $\phi$ is a classification-calibrated multi-class loss [43, 6]. Using this result, we can directly make use of any statistically valid surrogate loss in ordinary classification with a simple modification.

## 2.3 Problems with Probability Estimation for L2D

While prior literature has established how to construct consistent surrogates for L2D, it is less well-known whether such surrogates can actually provide calibrated estimates of classifier and deferral probabilities. As mentioned before, we are also interested in the true probability of the correctness of our prediction, i.e., the value of $\boldsymbol{\eta}(\boldsymbol{x})$ (label being $Y$) and $\Pr(M = Y | X = \boldsymbol{x})$ (expert is correct). To achieve this goal of probability estimation, the commonly used method is to combine the obtained scoring function $\boldsymbol{g}^*$ with a transformation function $\psi$ to make the composite function $\psi \circ \boldsymbol{g}^*$ a probability estimator $\hat{\boldsymbol{\eta}}^\psi(\boldsymbol{g}(\boldsymbol{x}))$. For example, the softmax function is frequently used as $\psi$ in ordinary multi-class classification to map the scoring function from $\mathbb{R}^K$ to $\Delta^K$. So far, we have not discussed whether these probability estimates are valid.

In the task of L2D for classification, we aim to estimate both $\boldsymbol{\eta}(\boldsymbol{x}) \in \Delta^K$ and $\Pr(M = Y | X = \boldsymbol{x}) \in [0, 1]$ with a $K + 1$-dimensional estimator, where its $y$-th dimension is the estimate of $\eta_y(\boldsymbol{x})$ for $y \in \mathcal{Y}$ and $K + 1$-th dimension is the estimate of $\Pr(M = Y | X = \boldsymbol{x})$. Correspondingly, the transformation $\psi$ should be from $\mathbb{R}^{K+1}$ to a range $\mathcal{P}$ that contains $\Delta^K \times [0, 1]$. It was shown in Theorem 1 of Mozannar and Sontag [30] that a probability estimator $\hat{\boldsymbol{\eta}}^{\mathrm{sm}}$ with the softmax output of scorer function $\boldsymbol{g}$ and an extra fractional transformation guarantees to recover class probabilities and expert accuracy at $\boldsymbol{g}^* \in \operatorname{argmin}_{\boldsymbol{g}} R_{L_{CE}}(\boldsymbol{g})$, which is formulated as:

$$\hat{\eta}_y^{\mathrm{sm}}(\boldsymbol{g}(\boldsymbol{x})) = \psi_y^{\mathrm{sm}}(\boldsymbol{g}(\boldsymbol{x})) / \left( 1 - \psi_{K+1}^{\mathrm{sm}}(\boldsymbol{g}(\boldsymbol{x})) \right), \quad \forall y \in \mathcal{Y} \cup \{K+1\}, \tag{5}$$

It is noticeable that the range of this estimator is $\hat{\boldsymbol{\eta}}^{\mathrm{sm}} \in \Delta^K \times [0, +\infty]$: the estimate of expert accuracy $\hat{\eta}_{K+1}^{sm}$ is **unbounded above** and will approach $+\infty$ if $\psi_{K+1}^{\mathrm{sm}}(\boldsymbol{u}) \to 1$. Verma and Nalisnick [44] pointed out that the unboundedness can hurt the performance of this probability estimator: if $\psi_{K+1}^{\mathrm{sm}}(\boldsymbol{g}(\boldsymbol{x})) > 1/2$, the estimated expert accuracy will be greater than 1 and thus meaningless. Experimental results also show the frequent occurrence of such meaningless results due to the overconfidence of deep models [19]. We further illustrate such miscalibration in Figure 2.

To mitigate this problem, a new OvA-based surrogate loss that can induce a bounded probability estimator while remaining consistent is proposed in [44], which has the following formulation:

$$L_{\mathrm{OvA}}(\boldsymbol{g}(\boldsymbol{x}), y, m) = \xi(g_y(\boldsymbol{x})) + \sum_{y' \neq y}^{K+1} \xi(-g_{y'}(\boldsymbol{x})) + [\![ m = y ]\!] (\xi(g_{K+1}(\boldsymbol{x})) - \xi(-g_{K+1}(\boldsymbol{x}))), \tag{6}$$

where $\xi$ is a binary proper composite [46, 40] loss. Its induced probability estimator $\hat{\boldsymbol{\eta}}^{\mathrm{OvA}}$ is:

$$\hat{\eta}_y^{\mathrm{OvA}}(\boldsymbol{g}(\boldsymbol{x})) = \psi_\xi(g_y(\boldsymbol{x})), \quad \forall y \in \mathcal{Y} \cup \{K+1\}, \tag{7}$$

where $\psi_\xi$ is a mapping to $[0, 1]$ determined by the binary loss, which makes $\hat{\boldsymbol{\eta}}^{\mathrm{OvA}}$ bounded. It is also experimentally shown that the $\hat{\boldsymbol{\eta}}^{\mathrm{OvA}}$ outperformed $\hat{\boldsymbol{\eta}}^{\mathrm{sm}}$ [30] in the task of probability estimation.

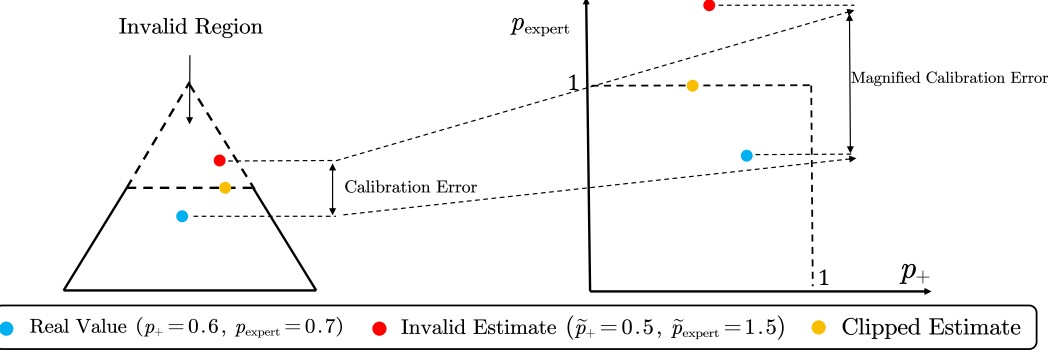

**Figure 2:** Illustration of the miscalibration of estimator (5) on a binary classification with deferral problem. The unbounded estimator (5) first estimates $\frac{[\boldsymbol{\eta}(\boldsymbol{x});\Pr(M=Y|X=\boldsymbol{x})]}{1+\Pr(M=Y|X=\boldsymbol{x})}$ that takes value in the 2-D probability simplex denoted by the left triangle and then obtain the final estimate with the fractional transformation (5). However, when over-confidence occurs in the 2-D simplex, i.e., the softmax output lies in the invalid region that $\psi_3^{\mathrm{sm}} > 1/2$, (5) will magnify the calibration error of the estimates and clipping it to a valid estimate cannot solve this problem.

Given the success of the OvA-based loss, can we assert that softmax parameterization is inferior to the OvA strategy in probability estimation for L2D? We argue in this paper that such a conclusion should not be drawn so quickly. We should not discourage the use of softmax parameterization based solely on the specific implementation (3), since there may be other implementations of softmax that can resolve the issues with $\hat{\boldsymbol{\eta}}^{\mathrm{sm}}$. Furthermore, the great success of softmax parameterization in deep learning models suggests that this potential implementation could also achieve outstanding performance. In this paper, we focus on finding such an implementation and show its superiority both theoretically and experimentally.

## 3 Problem with Symmetric Losses for L2D with Probability Estimation

Before beginning the search for a suitable implementation of softmax parameterization for L2D, it is important to determine the cause of the unboundedness of $\hat{\boldsymbol{\eta}}^{\mathrm{sm}}$. Is it due to the use of softmax parameterization or some other factor? If the reason is the former one, any attempts to create a bounded probability estimator with a softmax function for L2D will be in vain. Therefore, it is crucial to identify the root cause of the unboundedness before proceeding with further efforts.

Surprisingly, we find that such unboundedness is not a particular problem of softmax parameterization and is a common one shared by many loss functions. Recall that the loss (3) with unbounded estimator $\hat{\boldsymbol{\eta}}^{\mathrm{sm}}$ is a special case of the consistent surrogate (4) by setting the multi-class loss $\phi$ to softmax cross-entropy loss. We show that even if we use other losses beyond the softmax function and choose other losses such as the standard OvA losses ((14) in Zhang [52]), the induced probability estimators are still inevitably unbounded (or bounded but induced from an unbounded one) as long as we are using $\phi$ with symmetric structure:

**Theorem 1.** (Impossibility of non-trivial bounded probability estimator with symmetric losses)
If a surrogate loss $L_\phi$ defined as (4) has probability estimators and is induced from a symmetric consistent multi-class loss $\phi$ such that that $P\phi(\boldsymbol{u}) = \phi(P\boldsymbol{u})$, where $P$ is any permutation matrix, it must have an unbounded probability estimator $\hat{\boldsymbol{\eta}}$: let $\boldsymbol{g}^* \in \operatorname{argmin} R_{L_\phi}(\boldsymbol{g})$, we have that $\forall \hat{\boldsymbol{\eta}}(\cdot)$ where $\hat{\boldsymbol{\eta}}(\boldsymbol{g}^*(\boldsymbol{x})) = [\boldsymbol{\eta}(\boldsymbol{x}); \Pr(M = Y|X = \boldsymbol{x})]$, $\hat{\boldsymbol{\eta}}$ is not bounded above. Furthermore, any bounded probability estimator $\hat{\boldsymbol{\eta}}'$ for $L_\phi$ must be a piecewise modification of the unbounded estimator $\hat{\boldsymbol{\eta}}$: denote by $\mathcal{U} = \cup_{\boldsymbol{\beta}} \operatorname{argmin}_{\boldsymbol{u}} \sum_{y=1}^{K+1} \beta_y L_\phi(\boldsymbol{u}, y)$ for all $\boldsymbol{\beta} \in \Delta^K \times [0, 1]$, we have that $\hat{\boldsymbol{\eta}}'$ is equal to $\hat{\boldsymbol{\eta}}$ on $\mathcal{U}$.

The proof is provided in Appendix A. Intuitively, the existence of such an unbounded estimator is caused by the fact that symmetric loss induced $L_\phi$ neglects the asymmetric structure of the probabilities to be estimated by modeling $\frac{[\boldsymbol{\eta}(\boldsymbol{x});\Pr(M=Y|X=\boldsymbol{x})]}{1+\Pr(M=Y|X=\boldsymbol{x})}$ directly. Furthermore, all potential bounded probability estimators are piecewise modifications of the unbounded estimator sharing the same values on $\mathcal{U}$ with $\hat{\boldsymbol{\eta}}$, which means that they are generated by substituting the invalid values of the unbounded estimator with a valid one, e.g., we can obtain a bounded estimator based on the unbounded one (5) by clipping it to 1 if its estimated expert accuracy is larger than 1.

However, such modifications to make an unbounded estimator bounded are not very useful. Due to the training process's complexity and the distribution's arbitrariness, it is hard to design an appropriate modification of an unbounded estimator. For example, though we can bound the expert accuracy of (5) by 1 when it generates an invalid value that is larger than 1, it is still an overly confident estimation. Meanwhile, we do not know how to upper bound it by a value lower than 1 without prior knowledge of $\Pr(M = Y | X = \boldsymbol{x})$. Overall, training with $L_\phi$ in Theorem 1 can easily lead to a model that generates invalid probability estimation with an unbounded estimator $\hat{\boldsymbol{\eta}}$, while most modified bounded estimators cannot efficiently solve this problem. We will experimentally verify this point in Section 5.

This result shows that the consistent surrogate framework [9] does not directly provide us a solution for solving the unboundedness problem since most multi-class losses we use, e.g., CE loss, OvA loss, Focal loss [26], are symmetric. This motivates the design of surrogates with bounded probability estimators for L2D in our work. Based on Theorem 1, we have the following findings: firstly unboundedness of probability estimators is not necessarily caused by the softmax parameterization, as the induced estimator of any symmetric loss, e.g., standard softmax-free symmetric OvA losses [52], exhibits unboundedness too. Secondly, due to the fact that the OvA strategy for L2D (6) successfully gets rid of the unboundedness issue, we can reasonably deduce that a modified softmax function can also present similar outcomes. In the following section, we propose a novel modification of the softmax parameterization that overcomes these issues and reveals that both the previous OvA-based work [44] and our current work benefit from the use of asymmetric losses.

## 4 Consistent Softmax-Based Surrogate with Bounded Probability Estimator

In this section, we find a softmax parameterization that is a feasible solution in L2D for the construction of surrogate loss with a bounded probability estimator. We first propose a cross-entropy-like surrogate loss but with an asymmetric transformation that resembles the ordinary softmax function for its first $K$ elements but processes the $K + 1$-th coordinate in a special way. Then we show that such a transformation has useful properties for both classification and probability estimation, and then provide consistency analysis for both the loss function and its induced bounded probability estimator. Finally, we show that our proposed loss and the OvA loss [44] are connected to the consistent surrogate formulation (4) with the use of asymmetric multi-class losses, which indicates the helpfulness of asymmetry and shed light on the future study of consistent surrogates for L2D with bounded estimators.

### 4.1 Consistent Surrogate Formulation with Asymmetric Bounded Softmax Parameterization

Before searching for a softmax-parameterization that is capable of L2D with probability estimation, we need to better understand the failure of (3) for inducing a bounded probability estimator. By inspecting the proof of consistency in Theorem 1 of Mozannar and Sontag [30], we can learn that the cross-entropy-like surrogate (3) works by fitting a $K + 1$ class posterior distribution: $\psi^{\mathrm{sm}}(\boldsymbol{g}^*) = \tilde{\boldsymbol{\eta}}(\boldsymbol{x}) = [\frac{\eta_1(\boldsymbol{x})}{1+\Pr(M=Y|\boldsymbol{x})}, \cdots, \frac{\eta_K(\boldsymbol{x})}{1+\Pr(M=Y|\boldsymbol{x})}, \frac{\Pr(M=Y|\boldsymbol{x})}{1+\Pr(M=Y|\boldsymbol{x})}]$. Though we can check that the $\varphi \circ \boldsymbol{g}^*$ is exactly a Bayes optimal solution of $R_{01}^\perp(f)$ due to the monotonicity of the softmax function and the form of $\tilde{\boldsymbol{\eta}}$, to get $[\boldsymbol{\eta}(\boldsymbol{x}); \Pr(M = Y | X = \boldsymbol{x})]$ we need to perform an extra inverse transformation should be exerted on $\psi^{\mathrm{sm}}$. This is caused by the following dichotomy: the class probabilities and expert accuracy we aim to estimate are in the range of $\Delta^K \times [0, 1]$, while the standard softmax-parameterization of (3) maps the $K + 1$ dimensional scoring function $\boldsymbol{g}$ into $\Delta^{K+1}$.

To solve this issue, a promising approach is to modify the softmax function to make it a transformation that can directly cast the scoring function $\boldsymbol{g}$ into the target range $\Delta^K \times [0, 1]$. Since the target range is not symmetric, the modified softmax should also be asymmetric. This idea is given a concrete form in the following asymmetric softmax parameterization:

**Definition 2.** (Asymmetric softmax parameterization) $\tilde{\psi}$ is called a *asymmetric softmax function* that for any $K > 1$ and $\boldsymbol{u} \in \mathbb{R}^{K+1}$:

$$\tilde{\psi}_y(\boldsymbol{u}) = \begin{cases} \frac{\exp(u_y)}{\sum_{y'=1}^{K} \exp(u_{y'})}, & y \neq K + 1, \\ \frac{\exp(u_{K+1})}{\sum_{y'=1}^{K+1} \exp(u_{y'}) - \max_{y' \in \{1, \cdots, K\}} \exp(u_{y'})}, & \text{else.} \end{cases} \tag{8}$$

At first glance, the proposed asymmetric function appears to be the standard softmax function, which excludes the $K + 1$-th input of $\boldsymbol{u}$ for $y \neq K + 1$. However, the last coordinate of the function takes on a quite different form. This special asymmetric structure is designed to satisfy the following properties:

**Proposition 1.** (Properties of $\tilde{\psi}$) For any $\boldsymbol{u} \in \mathbb{R}^{K+1}$:
(i). (Boundedness) $\tilde{\psi}(\boldsymbol{u}) \in \Delta^K \times [0, 1]$,
(ii). (Maxima-preserving) $\operatorname{argmax}_{y \in \{1, \cdots, K+1\}} \tilde{\psi}_y(\boldsymbol{u}) = \operatorname{argmax}_{y \in \{1, \cdots, K+1\}} u_y$.

It can be seen that the proposed asymmetric softmax $\tilde{\psi}$ is not only bounded but also successfully maps $\boldsymbol{g}$ into the desired target range, which indicates that $\tilde{\psi}$ may directly serve as the probability estimator $\hat{\boldsymbol{\eta}}^{\tilde{\psi}}(\boldsymbol{g}(\boldsymbol{x})) = \tilde{\psi}(\boldsymbol{g}(\boldsymbol{x}))$. Furthermore, the maxima-preserving property guarantees that the estimator is also capable of discriminative prediction: if a scoring function $\boldsymbol{g}'$ can recover the true probability, i.e., $\hat{\boldsymbol{\eta}}^{\tilde{\psi}}(\boldsymbol{g}'(\boldsymbol{x})) = [\boldsymbol{\eta}(\boldsymbol{x}); \Pr(M = Y | X = \boldsymbol{x})]$, then $\varphi \circ \boldsymbol{g}'$ must also be the Bayes optimal solution of $R_{01}^{\perp}(\boldsymbol{g})$. Based on the asymmetric softmax function, we propose the following surrogate loss for L2D:

**Definition 3.** (Asymmetric Softmax-Parameterized Loss) The proposed surrogate for L2D with asymmetric softmax parameterization $L_{\tilde{\psi}}(\boldsymbol{u}, y, m) : \mathbb{R}^{K+1} \times \mathcal{Y} \times \mathcal{Y}$ is formulated as:

$$L_{\tilde{\psi}}(\boldsymbol{u}, y, m) = -\log(\tilde{\psi}_y(\boldsymbol{u})) - [\![m \neq y]\!] \log(1 - \tilde{\psi}_{K+1}(\boldsymbol{u})) - [\![m = y]\!] \log(\tilde{\psi}_{K+1}(\boldsymbol{u})). \quad (9)$$

The proposed loss takes an intuitive form, which can be seen as the combination of the cross-entropy loss (the first term) and a modified version of binary logistic loss (the last two terms). This formulation is inspired by the structure of our problem, where the expert accuracy is not directly related to class probabilities while the class probabilities should fulfill that $\boldsymbol{\eta}(\boldsymbol{x}) \in \Delta^K$. In fact, the proposed surrogate is not a simple summation of two independent losses and they are indirectly related by the asymmetric softmax function: according to Definition 2, the two counterparts share the same elements $[g_1, \cdots, g_K]$. This correlation serves as a normalization for $\tilde{\psi}_{K+1}$ that brings the property of maxima-preserving, which is crucial for the following consistency analysis:

**Theorem 2.** (Consistency of $L_{\tilde{\psi}}$ and bounded probability estimator $\hat{\boldsymbol{\eta}}^{\tilde{\psi}}$)
The proposed surrogate $L_{\tilde{\psi}}$ is a consistent surrogate for L2D, i.e., $\varphi \circ \boldsymbol{g}^* \in \operatorname{argmin}_f R_{01}^{\perp}(f)$, $\forall \boldsymbol{g}^* \in \operatorname{argmin}_{\boldsymbol{g}} R_{\Phi}^{\perp}(\boldsymbol{g})$. The bounded probability estimator can also recover the desired class probabilities and expert accuracy: $\hat{\boldsymbol{\eta}}^{\tilde{\psi}}(\boldsymbol{g}^*(\boldsymbol{x})) = [\boldsymbol{\eta}(\boldsymbol{x}); \Pr(M = Y | X = \boldsymbol{x})]$, $\forall \boldsymbol{x} \in \mathcal{X}$. Furthermore, if there exists other probability estimators for $L_{\tilde{\psi}}$, they must also be bounded.

The proof can be found in Appendix C. According to the theorem above, we showed that our proposed asymmetric softmax function induces a consistent surrogate and a bounded probability estimator. We also showed that there does not exist an unbounded probability estimator for $L_{\tilde{\psi}}$, which guarantees that our proposed bounded estimator is never the modification of an unbounded one. This result enriches the toolbox for L2D with probability estimation and theoretically justifies the use of softmax parameterization in L2D. In fact, we can further substitute the cross-entropy-like counterpart with any strictly proper loss [46] to get the same consistency result. In the following subsection, we will discuss the relationship between our method and the general consistent surrogate framework [9].

### 4.2 Connection with Consistent Surrogate Framework [9]: Asymmetry Can Help

In the previous section, we showed that there exists an implementation of softmax $\tilde{\psi}$ that can induce both a consistent loss and a valid probability estimator for L2D. Given the theoretical successes of our proposed softmax-based method and the previous OvA strategy, we may further expect them to provide more insights into the design of surrogates and probability estimators for L2D. Recalling the consistent surrogate formulation (4) proposed in Charusaie et al. [9] that allows the use of all consistent multi-class losses for constructing L2D surrogates, it is an instinctive idea that our proposed consistent surrogates (9) and the previous work [44] can be included in this framework. However, this idea may not be easily confirmed: Theorem 1 implies that the two surrogates are not the trivial combinations of commonly used symmetric losses and formulation (4) since they can both induce bounded probability estimators. The following corollary shows that the two surrogates are included in

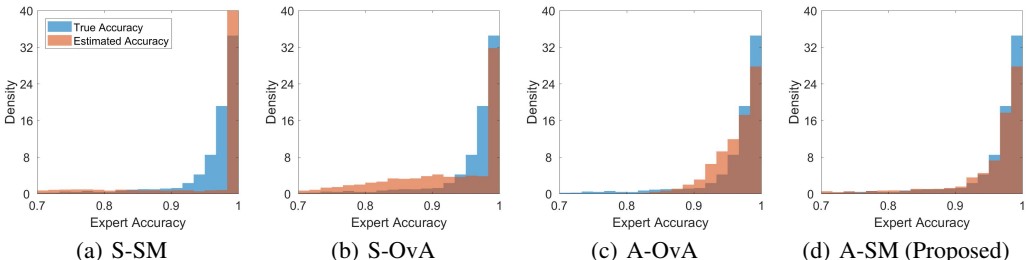

(a) S-SM        (b) S-OvA        (c) A-OvA        (d) A-SM (Proposed)

**Figure 3:** Distributions of the true and estimated accuracy of baselines and proposed method on CIFAR10H.

the family of consistent surrogate formulation (4), but induced from two novel asymmetric multi-class surrogates.

**Corollary 1.** We can get the consistent surrogates $L_{\tilde{\psi}}$ (9) and $L_{\text{OvA}}$ (6) by setting $\phi$ in the consistent loss formulation (4) to specific **consistent and asymmetric** multi-class losses.

Due to page limitations, we present the formulation of the asymmetric multi-class losses and the proof in the Appendix D. Although this conclusion reveals the connection between the two consistent surrogates and the general framework (4), it is important to note that it does not necessarily imply that the results in Verma and Nalisnick [44] and this paper are trivial. Since asymmetric losses are seldom used in ordinary multi-class classification, it is hard to obtain $L_{\tilde{\phi}}$ and $L_{\text{OvA}}$ by simply selecting $\phi$ from known consistent multi-class losses. According to this corollary and theorem 1, we can determine that a multi-class loss $\phi$ with asymmetric structure is a necessity if we aim to design consistent L2D surrogates with bounded probability estimators based on the general formulation (4). Based on this insight, it is promising to discover more surrogates with bounded probability estimators for L2D by focusing on the design of asymmetric multi-class surrogates.

### 4.3 Regret Transfer Bounds

In this section, we further study the regret transfer bounds of our proposed surrogate (9) to characterize the effect of minimizing surrogate risk $R_{L_{\tilde{\psi}}}(\boldsymbol{g})$ on both the target risk $R_{01}^{\perp}(\boldsymbol{g})$ and the misclassification error of the classifier. Denoted by $\boldsymbol{g}_{\text{class}} = \boldsymbol{g}_{1:K}$ the classifier counterpart of our model and $R_{01}(\boldsymbol{g})$ is its misclassification error, we can show that:

**Theorem 3.** $\max\left(R_{01}(\boldsymbol{g}_{\text{class}}) - R_{01}^*, \ R_{01}^{\perp}(\boldsymbol{g}) - R_{01}^{\perp *}\right) \leq \sqrt{2\left(R_{L_{\tilde{\psi}}}(\boldsymbol{g}) - R_{L_{\tilde{\psi}}}^*\right)}.$

The proof can be found in Appendix E. This theorem shows that we can use the excess risk of our proposed surrogate to upper-bound those of our concerned targets, and its proof is conducted by applying Pinsker's inequality and zooming.

This theorem shows that as long as we reach a low risk *w.r.t.* the proposed surrogate (approximately optimal), we can obtain a model with satisfying performances for both L2D and classification. This conclusion is not trivial: according to the characterization of $R_{01}^{\perp}$'s Bayes optimality in Definition 1, the classifier counterpart is not guaranteed to classify accurately on the deferred samples. We will further experimentally demonstrate the efficacy of our surrogate for L2D and classification in the next section.

## 5 Experiments

In this section, we compare our proposed asymmetric softmax estimator and its induced surrogate loss with the estimators and losses in previous works. Detailed setup can be found in Appendix F.

**Datasets and Models.** We evaluate the proposed method and baselines on widely used benchmarks with both synthetic and real-world experts. For synthetic experts, we conduct experiments on the CIFAR100 [24]. Following the previous works, the expert has a chance of $p$ to generate correct labels on the first $k \in \{20, 40, 60\}$ classes and at random otherwise. We set $p = 94\%$ as in Mozannar and Sontag [30] and $75\%$ as in Verma and Nalisnick [44] to simulate experts of high and medium accuracy. For real-world experts, we use CIFAR10H, HateSpeech, and ImageNet-16H [36, 14, 21]

**Table 1:** Experimental results on CIFAR100 with for 5 trials. We report the mean(%)(standard error(%)) of related statistics. The best and comparable methods based on t-test at significance level $5\%$ are highlighted in boldface.

| Method | Expert | Error | Coverage | ECE | Budgeted Error | | |
|---|---|---|---|---|---|---|---|
| | | | | | 10% | 20% | 30% |
| Expert Accuracy = 94% | | | | | | | |
| S-SM | 20 | **24.58(0.13)** | 77.72(0.31) | 4.86(0.11) | 31.68(0.39) | 25.00(0.24) | 24.59(0.13) |
| | 40 | **21.92(0.32)** | 57.89(0.54) | 9.22(0.38) | 46.69(0.63) | 38.50(0.60) | 29.94(0.62) |
| | 60 | **18.69(0.88)** | 40.34(6.28) | 11.10(0.72) | 59.44(4.95) | 51.36(5.08) | 42.77(5.09) |
| S-OvA | 20 | 27.00(1.91) | 85.40(0.91) | 5.15(0.31) | 28.42(2.26) | 27.00(1.91) | 27.00(1.91) |
| | 40 | 27.33(1.50) | 68.47(1.59) | 9.38(0.46) | 39.95(2.35) | 32.93(2.30) | 27.92(2.06) |
| | 60 | **18.44(0.64)** | 58.18(1.55) | 7.70(0.24) | 42.52(1.74) | 33.67(1.72) | 25.74(1.75) |
| A-OvA | 20 | 25.63(0.97) | 90.40(0.89) | **4.35(0.29)** | 25.65(0.99) | 25.63(0.97) | 25.63(0.97) |
| | 40 | 23.23(0.42) | 80.46(0.51) | **6.81(0.33)** | 27.44(0.62) | 23.23(0.42) | 23.23(0.42) |
| | 60 | **19.64(1.10)** | 70.44(2.73) | 7.34(0.38) | 31.90(2.73) | 24.75(2.73) | 20.05(1.62) |
| A-SM (Proposed) | 20 | **24.54(0.06)** | **98.16(0.03)** | 4.63(0.11) | **24.54(0.06)** | **24.54(0.06)** | **24.54(0.06)** |
| | 40 | **22.17(0.36)** | **92.20(0.32)** | 6.58(0.22) | **22.17(0.36)** | **22.17(0.36)** | **22.17(0.36)** |
| | 60 | **19.30(0.58)** | **84.72(0.30)** | 5.96(0.44) | **22.91(0.42)** | **19.30(0.58)** | **19.30(0.58)** |
| Expert Accuracy = 75% | | | | | | | |
| S-SM | 20 | 25.49(0.20) | 86.46(0.52) | 5.07(0.21) | 26.09(0.23) | 25.48(0.20) | 25.49(0.20) |
| | 40 | 25.13(0.44) | 74.65(0.48) | 12.82(0.29) | 32.82(0.66) | 26.99(0.59) | 25.13(0.44) |
| | 60 | 24.05(0.43) | 60.25(1.21) | 18.70(0.55) | 42.30(0.98) | 36.18(0.99) | 29.47(1.10) |
| S-OvA | 20 | 26.53(0.86) | 90.06(2.92) | 5.57(0.54) | 26.83(1.20) | 26.53(0.86) | 26.53(0.86) |
| | 40 | 26.89(1.78) | 77.69(5.82) | 7.14(1.41) | 32.30(4.92) | 28.33(3.11) | 26.90(1.78) |
| | 60 | 25.29(0.69) | 69.08(1.35) | 7.47(0.12) | 36.67(1.45) | 30.44(1.33) | 25.70(1.15) |
| A-OvA | 20 | 25.94(0.52) | 93.05(0.43) | **4.78(0.29)** | 25.94(0.52) | 25.94(0.52) | 25.94(0.52) |
| | 40 | 25.73(1.07) | 83.29(2.19) | **5.75(0.80)** | 27.75(1.39) | 25.73(1.07) | 25.73(1.07) |
| | 60 | **22.81(0.52)** | 79.76(1.96) | 6.10(0.32) | 27.19(1.64) | 23.08(0.87) | **22.81(0.52)** |
| A-SM (Proposed) | 20 | **24.58(0.12)** | **98.90(0.75)** | 4.34(0.31) | **24.58(0.12)** | **24.58(0.12)** | **24.58(0.12)** |
| | 40 | **24.29(0.35)** | **96.53(0.21)** | 5.58(0.13) | **24.29(0.35)** | **24.29(0.35)** | **24.29(0.35)** |
| | 60 | **22.48(0.32)** | **92.63(0.33)** | 5.58(0.17) | **22.48(0.32)** | **22.48(0.32)** | **22.48(0.32)** |

datasets, where the expert's prediction is generated using the provided auxiliary expert information. More experimental results of real-world experts can be found in Appendix F.

For CIFAR-100, HateSpeech, and ImageNet-16H, we report the misclassification error, coverage, and the ECE of the expert accuracy estimates. We also report the error of our model with the deferral budget as in Figure 4 of Verma and Nalisnick [44] to further evaluate the performance of the obtained classifier. For CIFAR10H, we plot the distribution of the true and estimated expert accuracy in Figure 3 to illustrate the performance of baselines and our method on fitting $\Pr(M = Y | X = \boldsymbol{x})$. The experiments on CIFAR-100 is conducted with 28-layer WideResNet [50] and SGD as in previous works [30, 44], and those on datasets with real-world experts are conducted using pre-trained models/embedding.

**Baselines.** We compare our **A**symmetric **S**oft**M**ax based method (A-SM) with the previously proposed **S**ymmetric **S**oft**M**ax-based (S-SM) method [30] and **A**symmetric OvA (A-OvA) based method [44]. We also combined the **S**ymmetric OvA logistic loss (S-OvA) with (4) to further evaluate the effect of symmetric loss and its induced unbounded estimator. For unbounded probability estimators in S-SM and S-OvA, we clip their estimates into $[0, 1]$ to make them valid. We directly use the output of baselines and our method without post-hoc techniques [19, 33] to better reflect their own performance.

**Experimental Results.** By observing Figure 3, we can find that the distributions of bounded method A-OvA and our proposed A-SM have markedly more overlap with the true distribution compared with the symmetric and unbounded ones, which directly shows that the bounded estimators can better estimate of expert accuracy. As can be seen from the experimental results reported in Table 1 and 2, our proposed A-SM is always better than or comparable to all the baselines *w.r.t.* classification accuracy. We can also see that the coverage of our method is always significantly higher than the baselines, which shows that our surrogate can induce ideal models for L2D. Though S-SM is comparable to A-SM with a high-accuracy expert, it has a significantly lower coverage and is

**Table 2:** Experimental results on HateSpeech and ImageNet-16H with noise type "095" for 5 trials. We report the mean(%)(standard error(%)) of related statistics.The best and comparable methods based on t-test at significance level 5% are highlighted in boldface.

| Method | Error | Coverage | ECE | Budgeted Error | | |
| --- | --- | --- | --- | --- | --- | --- |
| | | | | 10% | 20% | 30% |
| HateSpeech | | | | | | |
| S-SM | 8.65(0.14) | 70.19(1.50) | 3.95(0.48) | 25.48(0.49) | 16.80(0.44) | 8.97(0.33) |
| S-OvA | 8.65(0.14) | 69.90(0.53) | 1.77(0.07) | 24.19(0.25) | 15.89(0.34) | 8.78(0.31) |
| A-OvA | 9.64(0.32) | 77.2(0.42) | **1.73(0.26)** | 16.525(0.28) | 8.80(0.25) | 8.56(0.29) |
| A-SM | **8.06(0.40)** | **81.98(0.58)** | **1.53(0.26)** | **15.07(0.35)** | **8.06(0.40)** | **8.06(0.40)** |
| ImageNet-16H | | | | | | |
| S-SM | 15.08(1.19) | 26.41(1.78) | 22.92(2.59) | 68.58(1.30) | 60.33(1.19) | 51.27(1.94) |
| S-OvA | 15.15(1.50) | 23.08(2.39) | 11.68(0.53) | 68.91(2.51) | 60.08(1.82) | 51.50(2.05) |
| A-OvA | 14.17(0.58) | 21.56(2.13) | 11.57(1.50) | 71.14(2.10) | 62.39(2.10) | 53.95(2.27) |
| A-SM | **12.59(0.79)** | **39.48(2.50)** | **8.13(1.33)** | **57.75(1.37)** | **49.66(1.18)** | **40.85(1.51)** |

outperformed by A-SM with an expert of lower accuracy, which indicates that it is suffering from the problem of deferring more samples than necessary. This problem is also observed in learning with rejection [8]. Though S-OvA can mitigate this problem, its coverage is still consistently lower than A-SM. Notice that when there exist deferral budget requirements, all the unbounded baselines are outperformed by A-OvA, and A-OvA is further outperformed by A-SM, which indicates that our method can also efficiently generate a classifier. Meanwhile, the ECE of our estimated expert accuracy is also comparable to or better than A-OvA, while those of the unbounded ones all have significantly higher ECE, which further shows the efficacy of our method in estimating expert accuracy.

# 6 Conclusion

In this paper, we provide a novel consistent surrogate loss based on an asymmetric softmax function for learning to defer that can also provide calibrated probability estimates for the classifier and for expert correctness. We reveal that the root cause of the previous unbounded and miscalibrated probability estimators for L2D is not softmax but the intrinsic symmetry of the used loss function. We solve this problem by designing an asymmetric softmax function and using it to induce a consistent surrogate loss and a bounded probability estimator. We further give the regret transfer bounds of our method for both L2D and classification tasks. Finally, we evaluate our method and the baseline surrogate losses and probability estimators on benchmark datasets with both synthetic and real-world experts and show that we outperform prior methods. While we provide a consistent multi-expert extension of our proposed surrogate in Appendix H, it is still a promising future direction to generalize the multi-expert surrogates [45] to all the consistent multiclass losses as in Charusaie et al. [9].

# Acknowledgement

This research/project is supported by the National Research Foundation, Singapore and DSO National Laboratories under the AI Singapore Programme (AISG Award No: AISG2-GC-2023-009), and Ministry of Education, Singapore, under its Academic Research Fund Tier 1 (RG13/22). Lei Feng is supported by Chongqing Overseas Chinese Entrepreneurship and Innovation Support Program, CAAI-Huawei MindSpore Open Fund, and Openl Community (https://openi.pcl.ac.cn). Hussein Mozannar is thankful for the support of the MIT-IBM Watson AI Lab.

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

## A    Proof of Theorem 1

We first prove that if there is an L2D probability estimator $\hat{\boldsymbol{\eta}}$ for $L_\phi$, then we can reconstruct a probability estimator $\tilde{\boldsymbol{\eta}}$ for multiclass classification with $\phi$ if $\boldsymbol{\eta}$ is in $\tilde{\Delta}^{K+1}$, which is the collection of all the elements $\boldsymbol{\beta} \in \Delta^{K+1}$ with $\beta_{K+1} \leq \frac{1}{2}$. Then we show that according to the symmetry of $\phi$, we can extend this estimator to $\Delta^{K+1}$, and then construct an unbounded L2D probability estimator for $L_\phi$.

*Proof.* Denote by $R_{L_\phi}(\boldsymbol{u}, \boldsymbol{\eta}(\boldsymbol{x}), \Pr(M = Y|X = \boldsymbol{x})) = \sum_{y=1}^K \eta(\boldsymbol{x})_y \phi(\boldsymbol{u}, y) + \Pr(M = Y|X = \boldsymbol{x})\phi(\boldsymbol{u}, K+1)$ the conditional risk for L2D. Denote by $\hat{\boldsymbol{\eta}}$ a probability estimator for $L_\phi$ that $\hat{\boldsymbol{\eta}}(\boldsymbol{u}^*) = [\boldsymbol{\eta}(\boldsymbol{x}); \Pr(M = Y|X = \boldsymbol{x})]$ for all $\boldsymbol{u}^* \in \arg\min_{\boldsymbol{u}} R_{L_\phi}(\boldsymbol{u}, \boldsymbol{\eta}(\boldsymbol{x}), \Pr(M = Y|X = \boldsymbol{x}))$, we can learn that $\Delta^K \times [0, 1]$ is in the range of $\hat{\boldsymbol{\eta}}$. Then it is easy to verify that $\tilde{\boldsymbol{\eta}}(\boldsymbol{u}) = \frac{\hat{\boldsymbol{\eta}}(\boldsymbol{u})}{1+\hat{\eta}_{K+1}(\boldsymbol{u})} \in \tilde{\Delta}^{K+1}$ is a valid probability estimator for $K + 1$-class multiclass classification with $\phi$ if the posterior probabilities $[p(y|\boldsymbol{x})]_{y=1}^{K+1}$ is in $\tilde{\Delta}^{K+1}$.

Then we construct a valid estimator for $\Delta^{K+1}$ based on $\tilde{\boldsymbol{\eta}}$. Denote by $P$ a permutation matrix that exchanges the value of a vector's first and last dimension, then we have the following estimator $\tilde{\boldsymbol{\eta}}'$:

$$\tilde{\boldsymbol{\eta}}'(\boldsymbol{u}) = \begin{cases} \tilde{\boldsymbol{\eta}}(\boldsymbol{u}), & u_{K+1} \neq \max_y u_y, \\ P\tilde{\boldsymbol{\eta}}(P\boldsymbol{u}), & \text{else.} \end{cases}$$

We then prove that it is a valid estimator. Denote by $\Delta_+^{K+1}$ the set of all the $\boldsymbol{\beta} \in \Delta^{K+1}$ that $\beta_{K+1} \neq \max_y \beta_y$, we can learn that $\Delta_+^{K+1} \in \tilde{\Delta}^{K+1}$. Then for any $\boldsymbol{\beta} \in \Delta_+^{K+1}$, denote by $\boldsymbol{u}^*$ the minimizer of conditional risk *w.r.t.* $\boldsymbol{\beta}$ and $\phi$, we can learn that $u_{K+1}^* \neq \max_y u_y^*$ due to the **consistency** of $\phi$ and thus $\tilde{\boldsymbol{\eta}}'(\boldsymbol{u}^*) = \tilde{\boldsymbol{\eta}}(\boldsymbol{u}^*) = \boldsymbol{\beta}$.

For any $\boldsymbol{\beta} \notin \Delta_+^{K+1}$, we can learn that $P\boldsymbol{\beta} \in \Delta_+^{K+1}$. Denote by $\boldsymbol{u}^*$ the minimizer of conditional risk *w.r.t.* $\boldsymbol{\beta}$ and $\phi$, then we can learn that $P\boldsymbol{u}^*$ must be the minimizer of $P\boldsymbol{\beta}$ according to the **symmetry** of $\phi$. Then we have that $\tilde{\boldsymbol{\eta}}(P\boldsymbol{u}^*) = P\boldsymbol{\beta}$. Furthermore, notice that $PP\boldsymbol{\beta} = \boldsymbol{\beta}$, then we have $\tilde{\boldsymbol{\eta}}'(\boldsymbol{u}^*) = P\tilde{\boldsymbol{\eta}}(P\boldsymbol{u}^*) = PP\boldsymbol{\beta} = \boldsymbol{\beta}$.

Combining the two paragraphs above, we can learn that $\tilde{\boldsymbol{\eta}}'$ is a valid multiclass probability estimator for $\phi$. Then we can construct an unbounded L2D estimator as in (3), which indicates the existence of an unbounded probability estimator. Furthermore, since the loss function is unchanged, the collection of all the minimizers of $R_{L_\phi}(\boldsymbol{u}, \boldsymbol{\eta}(\boldsymbol{x}), \Pr(M = Y|X = \boldsymbol{x}))$ are also unchanged, and thus the all the values should have the same value on $\mathcal{U}$. $\square$

## B    Proof of Proposition 1

*Proof.* The boundedness of the first $K$ dimensions is straightforward. The $K + 1$th dimension's boundedness can also be directly proved by reformulating it as

$$\frac{\exp(u_{K+1})}{\exp(u_{K+1})+\sum_{y'=1}^K \exp(u_{y'})-\max_{y' \in \{1,\cdots,K\}} \exp(u_{y'})}.$$

Then we begin to prove its maxima-preserving. Denote by $y_1 = \arg\max_{y \in \{1,\cdots,K\}} u_y$. It is easy to verify that $\tilde{\psi}_{y_1}(\boldsymbol{u}) > \tilde{\psi}_y(\boldsymbol{u})$ for any $y \in \{1, \cdots, K\}/\{y_1\}$ due to the property of softmax function. Then we focus on the relation between $\tilde{\psi}_{y_1}(\boldsymbol{u})$ and $\tilde{\psi}_{K+1}(\boldsymbol{u})$. We can learn that:

$$\frac{\exp(u_{K+1})}{\exp(u_{K+1})+\sum_{y'=1}^K \exp(u_{y'})-\exp(u_{y_1})} = \frac{1}{1+\frac{\sum_{y'=1}^K \exp(u_{y'})-\exp(u_{y_1})}{\exp(u_{K+1})}},$$

$$\frac{\exp(u_{y_1})}{\sum_{y'=1}^K \exp(u_{y'})} = \frac{1}{1+\frac{\sum_{y'=1}^K \exp(u_{y'})-\exp(u_{y_1})}{\exp(u_{y_1})}}.$$

When $u_{K+1} > u_{y_1}$, we can learn that $\frac{\sum_{y'=1}^K \exp(u_{y'})-\exp(u_{y_1})}{\exp(u_{K+1})} < \frac{\sum_{y'=1}^K \exp(u_{y'})-\exp(u_{y_1})}{\exp(u_{y_1})}$, then $\tilde{\psi}_{K+1}(\boldsymbol{u}) > \tilde{\psi}_{y_1}(\boldsymbol{u})$. Based on the formulations above, it is also easy to verify the cases when $u_{y_1} \geq u_{K+1}$. Combining the discussions above and we can conclude the proof. $\square$

## C   Proof of Theorem 2

*Proof.* The consistency can be directly obtained by recovering the class probability and expert accuracy according to the maxima-preserving, then we first focus on how our proposed estimator recovers the posterior probabilities.

According to the property of log loss, it is easy to learn that $\tilde{\psi}_y(\boldsymbol{g}^*(\boldsymbol{x})) = p(y|\boldsymbol{x})$ for $y \in \{1, \cdots, K\}$. When $y = K + 1$, we can learn from the range of our estimator and the property of binary log loss that $\tilde{\psi}_y(\boldsymbol{g}^*(\boldsymbol{x})) = \Pr(M = Y|X = \boldsymbol{x})$. Then we can conclude that $\tilde{\psi}(\boldsymbol{g}^*(\boldsymbol{x})) = \boldsymbol{\eta}(\boldsymbol{x}) \times \Pr(M = Y|X = \boldsymbol{x})$.

Then we begin to prove that there is no unbounded probability estimator by contradiction. Suppose there exists an unbounded estimator $\psi$. For any $\boldsymbol{g}$, it must be the solution of a distribution and expert whose posterior probability is $\tilde{\psi}(\boldsymbol{g}(\boldsymbol{x}))$ for each point $\boldsymbol{x}$. However, for a $\boldsymbol{g}$ that there exists $\boldsymbol{x}$ that $\psi(\boldsymbol{g}(\boldsymbol{x})) \notin \Delta^K \times [0, 1]$, we can learn that it cannot be the solution of any distribution and expert according to the definition of probability estimator. We can learn from this contradiction that $\psi$ is not a probability estimator as long as its range is not $\Delta^K \times [0, 1]$. $\qquad\square$

## D   Proof of Corollary 1

*Proof.* We can set the multi-class loss to $\phi_{\tilde{\psi}}$ to get our proposed loss:

$$
\phi_{\tilde{\psi}}(\boldsymbol{u}, y) = \begin{cases} -\log\left(\frac{\exp(u_y)}{\sum_{y'=1}^{K'-1}\exp(u_{y'})}\right) - \log\left(1 - \frac{\exp(u_{K'})}{\sum_{y'=1}^{K'}\exp(u_{y'}) - \max_{y' \in [1, K'-1]}\exp_{u_{y'}}}\right), \ y \neq K', \\ -\log\left(\frac{\exp(u_{K'})}{\sum_{y'=1}^{K'}\exp(u_{y'}) - \max_{y' \in [1, K'-1]}\exp_{u_{y'}}}\right) + \log\left(1 - \frac{\exp(u_{K'})}{\sum_{y'=1}^{K'}\exp(u_{y'}) - \max_{y' \in [1, K'-1]}\exp_{u_{y'}}}\right), \ \text{else}. \end{cases}
$$

A similar result can be deduced for the OvA-based surrogate by considering the following consistent multi-class loss with a strictly proper binary composite loss $\xi$:

$$
\phi_{\text{OvA}}(\boldsymbol{u}, y) = \begin{cases} \xi(u_y) + \sum_{y' \neq y}\xi(-u_{y'}), \ y \neq K', \\ \xi(u_{K'}) - \xi(-u_{K'}), \ \text{else}. \end{cases}
$$

We then begin to prove their consistency. It is easy to verify that $\phi_{\tilde{\psi}}$ is minimized when $\tilde{\psi}(\boldsymbol{g}(\boldsymbol{x})) = \frac{p(y|\boldsymbol{x})}{1 - p(K'|\boldsymbol{x})}$, and we can conclude its consistency using the maxima-preserving property of $\tilde{\psi}$.

For $\phi_{\text{OvA}}$, denote by $\psi_{\text{OvA}}$ the inverse link of $\xi$. We can learn that $\boldsymbol{g}^*(\boldsymbol{x}) = \psi_{\text{OvA}}(\frac{p(y|\boldsymbol{x})}{1 - p(K'|\boldsymbol{x})})$, and we can learn the consistency since $\xi$ is strictly proper and thus $\psi_{\text{OvA}}$ is increasing. $\qquad\square$

## E   Proof of Theorem 3

*Proof.* We first apply Pinsker's inequality. We can learn that:

$$
R_{L_{\tilde{\psi}}}(\boldsymbol{g}) - R^*_{L_{\tilde{\psi}}} \geq \mathbb{E}_{p(\boldsymbol{x})}\left[\frac{1}{2}\|\tilde{\psi}_{1:K}(\boldsymbol{g}(\boldsymbol{x})) - \boldsymbol{\eta}(x)\|_1^2 + 2(\tilde{\psi}_{K+1}(\boldsymbol{g}(\boldsymbol{x}) - \Pr(M = Y|X = \boldsymbol{x}))^2\right].
$$

We can learn that $R_{L_{\tilde{\psi}}}(\boldsymbol{g}) - R^*_{L_{\tilde{\psi}}} \geq \mathbb{E}_{p(\boldsymbol{x})}\left[\frac{1}{2}\|\tilde{\psi}_{1:K}(\boldsymbol{g}(\boldsymbol{x})) - \boldsymbol{\eta}(x)\|_1^2\right]$ immediately and learn that the bound for $R_{01}$ following the analysis of cross-entropy loss in ordinary classification. Then we move to analyze the 0-1-deferral risk. We can further learn that :

$$
R_{L_{\tilde{\psi}}}(\boldsymbol{g}) - R^*_{L_{\tilde{\psi}}} \geq \mathbb{E}_{p(\boldsymbol{x})}\left[\frac{1}{2}\|\tilde{\psi}_{1:K}(\boldsymbol{g}(\boldsymbol{x})) - \boldsymbol{\eta}(x)\|_1^2 + 2(\tilde{\psi}_{K+1}(\boldsymbol{g}(\boldsymbol{x})) - \Pr(M = Y|X = \boldsymbol{x}))^2\right]
$$

$$
\geq \frac{1}{2}\mathbb{E}_{p(\boldsymbol{x})}\left[\|\tilde{\psi}_{1:K}(\boldsymbol{g}(\boldsymbol{x})) - \boldsymbol{\eta}(x)\|_1^2 + (\tilde{\psi}_{K+1}(\boldsymbol{g}(\boldsymbol{x})) - \Pr(M = Y|X = \boldsymbol{x}))^2\right]
$$

For any $\boldsymbol{x}$, when $\boldsymbol{g}(\boldsymbol{x})$ can induce the Bayes optimal solution for it, the excess risk is zero and the bound holds naturally. When it is not optimal, denote by $\eta_{K+1}(\boldsymbol{x}) = \Pr(M = Y|X = \boldsymbol{x})$. $y'$ is the

**Table 2:** Experimental results on CIFAR10H with for 5 trials. We report the mean(%)(standard error(%)) of related statistics.The best and comparable methods based on t-test at significance level 5% are highlighted in boldface.

| Method | Error | Coverage | ECE | Budgeted Error | | |
| --- | --- | --- | --- | --- | --- | --- |
| | | | | 10% | 20% | 30% |
| S-SM | 3.82(0.02) | 41.28(0.14) | 6.64(0.22) | 50.25(0.57) | 41.00(0.81) | 31.47(0.37) |
| S-OvA | 4.30(0.12) | 37.08(1.36) | 2.46(0.48) | 52.75(0.40) | 43.31(0.58) | 33.23(0.72) |
| A-OvA | 4.00(0.11) | 92.75(0.24) | **0.95(0.16)** | 4.00(0.11) | 4.00(0.11) | 4.00(0.11) |
| A-SM | **3.65(0.09)** | **94.20(0.37)** | 0.94(0.17) | **3.65(0.09)** | **3.65(0.09)** | **3.65(0.09)** |

**Table 3:** Experimental results on ImageNet-16H with for 5 trials. We report the mean(%)(standard error(%)) of related statistics.The best and comparable methods based on t-test at significance level 5% are highlighted in boldface.

| Method | Error | Coverage | ECE | Budgeted Error | | |
| --- | --- | --- | --- | --- | --- | --- |
| | | | | 10% | 20% | 30% |
| Image Noise Type = "080" | | | | | | |
| S-SM | 10.63(1.34) | 22.59(1.59) | 18.31(3.22) | 66.58(3.91) | 60.41(2.98) | 53.33(2.79) |
| S-OvA | 10.48(0.64) | 21.88(2.75) | 12.79(0.33) | 70.50(3.78) | 61.58(4.64) | 51.22(3.37) |
| A-OvA | 10.36(1.09) | 25.86(3.56) | 8.69(0.63) | 70.83(1.25) | 61.32(1.70) | 52.26(0.95) |
| A-SM | **7.94(0.85)** | **37.32(2.72)** | **7.39(0.23)** | **54.24(2.89)** | **45.33(2.91)** | **34.11(1.65)** |

dimension with the largest value of $\boldsymbol{g}(\boldsymbol{x})$ and that of $\boldsymbol{\eta}(\boldsymbol{x})$ is $y''$. Then we can learn that:

$$\frac{1}{2}\left(\|\tilde{\psi}_{1:K}(\boldsymbol{g}(\boldsymbol{x})) - \boldsymbol{\eta}(x)\|_1^2 + (\tilde{\psi}_{K+1}(\boldsymbol{g}(\boldsymbol{x})) - \Pr(M = Y|X = \boldsymbol{x}))^2\right)$$
$$\geq \frac{1}{2}\left(\tilde{\psi}_{y'}(\boldsymbol{g}(\boldsymbol{x})) - \eta_{y'}(\boldsymbol{x}) - \tilde{\psi}_{y''}(\boldsymbol{g}(\boldsymbol{x})) + \eta_{y''}(\boldsymbol{x})\right)^2$$
$$\geq \frac{1}{2}\left(\eta_{y'}(\boldsymbol{x}) - \eta_{y''}(\boldsymbol{x})\right)^2$$

The last step is obtained according to the maxima-preserving property. Further generalizing $y'$ and $y''$ to be instance-dependent ($y'(\boldsymbol{x})$ and $y''(\boldsymbol{x})$), we can learn the following inequality using Jensen's inequality:

$$R_{L_{\tilde{\psi}}}(\boldsymbol{g}) - R_{L_{\tilde{\psi}}}^* \geq \frac{1}{2}(\mathbb{E}_{p(\boldsymbol{x})}[|\eta_{y'}(\boldsymbol{x}) - \eta_{y''}(\boldsymbol{x})|])^2,$$

which concludes the proof since the second expectation term is $R_{01}^\perp(\boldsymbol{g}) - R_{01}^{\perp*}$ $\qquad\square$

## F   Details of Experiments

**Details of Model and Optimizer:**   For methods on CIFAR-10, we use the 28-layer WideResNet that is the same as those used in Mozannar and Sontag [30], Charusaie et al. [9]. The optimizer is SGD with cosine annealing, where the learning rate is 1e-1 and weight decay is 5e-4. We conduct the experiments on 8 NVIDIA GeForce 3090 GPUs and the batch size is 1024 (128 on each GPU). The training epoch on CIFAR100 is set to 200.

For methods on CIFAR-10H, we train the same WideResNet on the fully-label CIFAR-10 dataset and then train the linear layer using the CIFAR-10H dataset. The optimizer is AdamW [] and the learning rate, batch size, epoch number are 1e-4, 128, 200. For methods on HateSpeech, we use a 384-dimension embedding and linear model in the experiment. The optimizer is SGD with cosine annealing and the learning rate, batch size, epoch number is 0.1, 128, 50. For methods on ImageNet-16H, we use the embedding generated by the pretrained DenseNet and linear model. The optimizer is Adam and the learning rate, batch size, epoch number is 1e-3, 1e-1, 100.

The extra result on ImageNet with noise type "080" and the exact statistics for CIFAR-10H are shown in the tables in the appendix.

**Details of Evaluation Metrics:**    The reported Error is the sample mean of $\ell_{01}^{\perp}$, and Coverage is the ratio of undeferred samples. The ECE of expert accuracy is defined below:

$$\text{ECE} = \sum_{i=1}^{N} b_i |p_i - c_i|,$$

where $b_i$ is the ratio of predictions whose confidences fall into the $i$th bin. $p_i$ is the average confidence and $c_i$ is the average accuracy in this bin. We set the bin number to 15. The budgeted error is calculated as below: if the coverage is lower than $1 - x\%$, we will use the classifier's prediction instead of the expert's for those samples whose estimated expert accuracy is lower to make the coverage equal to $1 - x\%$.

## G    Limitations and Broader Impact

**Limitations:**    This work is designed for L2D without extra constraints on the number of expert queries. We believe that combining it with selective learning, i.e., adding explicit constraints on the ratio of deferred samples, can be a promising future direction.

**Broader Impact:**    When applied in real-world applications, the frequency of expert queries may be imbalanced due to the performance differences of the expert among samples. This is a common impact shared by all the L2D methods. We believe that introducing fairness targets into L2D can be another promising direction.

## H    Consistent Multi-Expert Extension

We provide a direct extension of our loss for the multi-expert setting and provide consistency analysis: Denote by the expert advice $\boldsymbol{m} = [m_1, \cdots, m_M] \in \mathcal{Y} \times \cdots \times \mathcal{Y}$. We first define an extended ASM:

**Definition 4.** (Multi-expert asymmetric softmax parameterization) $\tilde{\psi}$ is called a *multi-expert asymmetric softmax function* that for any $K > 1$ and $\boldsymbol{u} \in \mathbb{R}^{K+M}$:

$$\tilde{\psi}_y^M(\boldsymbol{u}) = \begin{cases} \frac{\exp(u_y)}{\sum_{y'=1}^{K} \exp(u_{y'})}, & y \notin [K+1, K+M], \\ \frac{\exp(u_y)}{\sum_{y'=1}^{K+1} \exp(u_{y'}) + \exp(y) - \max_{y' \in \{1, \cdots, K\}} \exp(u_{y'})}, & \text{else.} \end{cases} \tag{10}$$

We can directly extend the property of single-expert ASM to this case:

**Proposition 2.** (Properties of $\tilde{\psi}^M$) For any $\boldsymbol{u} \in \mathbb{R}^{K+M}$:
(i). (Boundedness) $\tilde{\psi}^M(\boldsymbol{u}) \in \Delta^K \times [0,1]^M$,
(ii). (Maxima-preserving) $\text{argmax}_{y \in \{1, \cdots, K+M\}} \tilde{\psi}_y^M(\boldsymbol{u}) = \text{argmax}_{y \in \{1, \cdots, K+M\}} u_y$.

Then we give the following extension:

**Definition 5.** (Multi-expert extension)

$$L_{\tilde{\psi}^M}(\boldsymbol{u}, y, m) = -\log(\tilde{\psi}_y^M(\boldsymbol{u})) - \sum_{i=1}^{M} \left( [\![m_i \neq y]\!] \log(1 - \tilde{\psi}^M_{K+i}(\boldsymbol{u})) - [\![m_i = y]\!] \log(\tilde{\psi}^M_{K+i}(\boldsymbol{u})) \right).$$

According to the position, we can simply give the characterization of $L_{\tilde{\psi}^M}$'s optimal solution:

$$\tilde{\psi}_y^{M*}(\boldsymbol{u}) = \begin{cases} p(y|\boldsymbol{x}), & y \notin [K+1, K+M], \\ \Pr(M_{y-K}|\boldsymbol{x}), & \text{else.} \end{cases} \tag{11}$$

According to the maxima-preserving, we can directly learn the consistency of our multiclass extension.

# I   Estimation Error Bound

In this section, we give a routine analysis of our method's estimation error bound. Our proof of the estimation error bound is based on Rademacher complexity:

**Definition 6.** (Rademacher complexity) Let $Z_1, \ldots, Z_n$ be $n$ i.i.d. random variables drawn from a probability distribution $\mu, \mathcal{H} = \{h : \mathcal{Z} \to \mathbb{R}\}$ be a class of measurable functions. Then the expected Rademacher complexity of $\mathcal{H}$ is defined as

$$\mathfrak{R}_n(\mathcal{H}) = \mathbb{E}_{Z_1, \ldots, Z_n \sim \mu} \mathbb{E}_{\boldsymbol{\sigma}} \left[ \sup_{h \in \mathcal{H}} \frac{1}{n} \sum_{i=1}^n \sigma_i h(Z_i) \right]$$

where $\boldsymbol{\sigma} = \{\sigma_1, \ldots, \sigma_n\}$ are Rademacher variables taking the value from $\{-1, +1\}$ with even probabilities.

First, let $\mathcal{G} \subset \mathcal{X} \to \mathbb{R}^{K+1}$ be the model class and each of its dimension is constructed by $\mathcal{G}_y \subset \mathcal{X} \to \mathbb{R}$. Then, we define the following scoring function space for L2D task:

$$\mathcal{L} \circ \mathcal{G} = \{h : (\boldsymbol{x}, y, m) \mapsto L_{\tilde{\psi}}(\boldsymbol{g}(\boldsymbol{x}), y, m) | \boldsymbol{g} \in \mathcal{G}\}$$

So the Rademacher complexity of $\mathcal{L} \circ \mathcal{G}$ given $n$ i.i.d. samples drawn from distribution with density $p(\boldsymbol{x}, y, m)$ can be defined as

$$\mathfrak{R}_n(\mathcal{L} \circ \mathcal{G}) = \mathbb{E}_{p(\boldsymbol{x}, y, m)} \mathbb{E}_{\boldsymbol{\sigma}} \left[ \sup_{g \in \mathcal{G}} \frac{1}{n} \sum_{i=1}^n \sigma_i h(\boldsymbol{x}_i, y_i, m_i) \right].$$

Denote by $\hat{R}_{L_{\tilde{\psi}}}(\boldsymbol{g}) = \frac{1}{n} \sum_{i=1}^n L_{\tilde{\psi}}(\boldsymbol{g}(\boldsymbol{x}_i), y_i, m_i)$ the empirical risk and $\hat{g}$ the empirical risk minimizer. We have the following theorem:

**Lemma 1.**

$$R_{L_{\tilde{\psi}}}(\hat{\boldsymbol{g}}) - R_{L_{\tilde{\psi}}}(\boldsymbol{g}^*) \le 2 \sup_{\boldsymbol{g} \in \mathcal{G}} |\hat{R}_{L_{\tilde{\psi}}}(\boldsymbol{g}) - R_{L_{\tilde{\psi}}}(\boldsymbol{g})|$$

*Proof.*

$$\begin{aligned} R_{L_{\tilde{\psi}}}(\hat{\boldsymbol{g}}) - R_{L_{\tilde{\psi}}}(\boldsymbol{g}^*) &= R_{L_{\tilde{\psi}}}(\hat{\boldsymbol{g}}) - \hat{R}_{L_{\tilde{\psi}}}(\hat{\boldsymbol{g}}) + \hat{R}_{L_{\tilde{\psi}}}(\hat{\boldsymbol{g}}) - R_{L_{\tilde{\psi}}}(\boldsymbol{g}^*) \\ &\le R_{L_{\tilde{\psi}}}(\hat{\boldsymbol{g}}) - \hat{R}_{L_{\tilde{\psi}}}(\hat{\boldsymbol{g}}) + \hat{R}_{L_{\tilde{\psi}}}(\boldsymbol{g}^*) - R_{L_{\tilde{\psi}}}(\boldsymbol{g}^*) \\ &\le 2 \sup_{\boldsymbol{g} \in \mathcal{G}} \left| \hat{R}_{L_{\tilde{\psi}}}(\boldsymbol{g}) - R_{L_{\tilde{\psi}}}(\boldsymbol{g}) \right| \end{aligned}$$

which completes the proof. $\qquad\qquad\square$

Then it is routing to get the estimation error bound with Rademacher complexity and McDiarmid's inequality:

**Theorem 4.** Suppose $L_{\tilde{\psi}}(\boldsymbol{g}(\boldsymbol{x}), y, m) \le M$ and $L_{\tilde{\psi}}$ is $L-$Lipschitz *w.r.t.* $\boldsymbol{g}(\boldsymbol{x})$, then with probability at least $1 - \delta$:

$$R_{L_{\tilde{\psi}}}(\hat{\boldsymbol{g}}) - R_{L_{\tilde{\psi}}}(\boldsymbol{g}^*) \le 4\sqrt{2}L \sum_{y=1}^{K+1} \mathfrak{R}_n(\mathcal{G}_y) + 4M\sqrt{\frac{\ln(2/\delta)}{2n}}$$

*Proof.* We will only discuss a one-sided bound on $\sup_{\boldsymbol{g} \in \mathcal{G}} \left( \hat{R}(\boldsymbol{g}) - R(\boldsymbol{g}) \right)$ that holds with probability at least $1 - \frac{\delta}{2}$. Altering $(\boldsymbol{x}_{[i]}, y_i, m_i)$ to $(\boldsymbol{x}'_{[i]}, y'_i, m'_i)$ and we can get a perturbed empirical risk $\hat{R}'(\boldsymbol{g})$. We can learn that

$$|\sup_{\boldsymbol{g} \in \mathcal{G}} \left( \hat{R}'(\boldsymbol{g}) - R(\boldsymbol{g}) \right) - \sup_{\boldsymbol{g} \in \mathcal{G}} \left( \hat{R}(\boldsymbol{g}) - R(\boldsymbol{g}) \right)| \le |\sup_{\boldsymbol{g} \in \mathcal{G}} \left( \hat{R}'(\boldsymbol{g}) - \hat{R}(\boldsymbol{g}) \right)| \le \frac{M}{n}.$$

Then we can use Mcdiarmids' inequality and vector-valued Talagrand inequality to learn:

$$\sup_{\boldsymbol{g} \in \mathcal{G}} \left| \hat{R}_{L_{\tilde{\psi}}}(\boldsymbol{g}) - R_{L_{\tilde{\psi}}}(\boldsymbol{g}) \right| \le 2\sqrt{2}L \sum_{y=1}^{K+1} \mathfrak{R}_n(\mathcal{G}_y) + 2M\sqrt{\frac{\ln(2/\delta)}{2n}}$$

$\qquad\qquad\square$

Using the lemma 1 and we can conclude our proof.

