# OpenReview forum: "In Defense of Softmax Parametrization for Calibrated and Consistent Learning to Defer"
_NeurIPS.cc/2023/Conference — NeurIPS 2023 poster_

### Official Review · Reviewer_r3aW · 2023-07-01

**Soundness:** 4 excellent
**Presentation:** 3 good
**Contribution:** 2 fair
**Rating:** 6
**Confidence:** 3

**Summary:**

This paper considers the problem of learning to defer, and propose a softmax-based surrogate loss for the task, which is consistent and can be well-calibrated. The paper theoretically proves that one needs asymmetric losses for a bounded probability estimate---which in turn leads to a better calibration properties---(Theorem 1), and show the consistency of the proposed loss (Theorem 2). The paper also gives the risk transfer bound to the original zero-one loss (Theorem 3). Empirically, the proposed surrogate function enjoys the best of both worlds, achieving low error rates and good calibration properties.

**Strengths:**

- Theoretical contributions of the paper is very good. Theorem 1 is a strong result, which may be useful in guiding the design of other softmax-based estimators for nonclassic (e.g., composite) learning tasks. Other two are somewhat standard, but are very essential in guaranteeing that the proposed loss works as desired.

- The proposed asymmetric softmax-parameterized loss is very simply designed. The loss can be implemented very easily and may not introduce too much computational overhead to the training procedure.

- The empirical performance of the proposed loss function is strong, consistently achieving the best performance over the tested setups.

- The paper is very clearly written, especially for a paper that is heavily loaded with various notations.

**Weaknesses:**

- It is not clear **why** we want to stick to the softmax-based losses from the first place. It is definitely cool that one can give a better calibrated softmax-based loss, but what is the reason that we need one when we have a decent non-softmax loss? We only know post hoc that such estimator has some empirical benefits, but the reason why it works better than non-softmax one is still at mystery. In the manuscript, authors try to justify the effort saying "Given the wide use and practical advantages of softmax parameterization (...)," but the reason is never spoken out clearly. For a better understanding of the advantages of the proposed loss, we need a more detailed explanation (or maybe a good reference).

- I find that the empirical validation of the loss function to be somewhat limited. I don't usually ask for more experimental validations for theory papers. However, I think it is quite essential for this paper specifically, because none of the theoretical results actually explain why the proposed loss function should work better than the OvA loss. The main experiments are on the synthetic CIFAR100 dataset, with an additional figure on CIFAR10H (where are the "error rate" results for CIFAR10H?). This is very limited; [42] performed experiments on CIFAR10H, Hatespeech, Galaxy-Zoo, and HAM10000; [28] has experiments on CIFAR10H, CheXpert, and Hatespeech. In this sense, having only CIFAR100 results stated explicitly as numbers seems rather inadequate, as we cannot really check whether the baselines have been implemented/tuned correctly.



**Questions:**

- What is the reason we should prefer softmax-based losses?

- Please provide more experimental results on other datasets.

- Please provide explicit numbers for the CIFAR10H experiments; this is for the sake of checking whether the baselines have been implemented and tuned correctly.

**Limitations:**

The limitations are adequately stated in the manuscript, but only in the appendices.

---

> ### Author Rebuttal · Authors · 2023-08-10
>
> **Q1. Please provide more experimental results on other datasets.**
>
> A1. Thank you for your constructive advice! We have added more experiments on datasets with real-world expert annotations, including Hatespeech [1] and ImageNet-16H [2] (2 tasks: “080” and “095”). The experimental results are provided in the one-page attachment. According to the results, the performance of our proposed surrogate continues to be comparable to or better than the baselines.
>
> The detailed experimental setup is provided below. For the model used for Hatespeech and ImageNet-16H, we apply the settings in [3]. The optimizer for Hatespeech is SGD + Cosine annealing is used for all the method for 50 epochs, and the learning rate, momentum, and batch size is 0.1, 0.9, and 1000. For both of the tasks in ImageNet-16H, the optimizer is Adam and the learning rate, batch size, and epoch number are set to be 1e-3, 100, and 100. Training-validation-testing split is set to 7:1:2 for both datasets.
>
> **Q2. Please provide explicit numbers for the CIFAR10H experiments; this is for the sake of checking whether the baselines have been implemented and tuned correctly.**
>
> A2. Thank you for raising this concern! In this main body, we conducted experiments with data from CIFAR-10H only to visualize the result of expert accuracy estimation, and thus the accuracy for all the methods is around 50%, which is far from optimal. To better reflect the ability of expert accuracy estimation of all the methods and evaluate the other abilities at the same time, we conduct additional experiments with both data from CIFAR-10H and CIFAR-10 as in [3]. The new visualization and a table of experimental results are provided in the one-page attachment. The experimental results show that our proposed method still outperformed the baselines. While the new visualization of probability estimation of S-OVA, A-OVA, and A-SM (proposed) remains similar to those in the main body, it can be seen that the estimation of S-SM severely suffers from the problem of unbounded confidence, which again shows the necessity of the study of bounded estimators.
>
> The detail of the experimental setup is listed below. The choice of model is the same as in [4], while the optimizer is SGD + Cosine annealing. The learning rate, momentum, batch size, and epoch number is 0.1, 0.9, 128, and 200. Training-validation-testing split is 7:1:2.
>
> **Q3. What is the reason we should prefer softmax-based losses?**
>
> A3. Thank you for raising this concern! In the task of ordinary multi-class classification, softmax-based loss functions, e.g., softmax cross-entropy loss, are the most widely used losses to train DNNs due to its superb performance. This is because the softmax parameterization can naturally model the K-dimensional probability simplex. However, the methods based on binary classification reduction, e.g., one-versus-all, fail to achieve this property. According to the resemblance between the task of multi-class classification and learning to defer, it is reasonable to infer that softmax-based losses could persist its advantage in L2D if implemented properly. The additional experimental results also strengthen the validity of this reason. As a result, we conclude that in the field of L2D and its probability estimation, the softmax-based losses should not be discouraged, and it is encouraged to give priority to the use of it as in the multi-class classification task.
>
> [1]. Davidson, T., Warmsley, D., Macy, M., and Weber, I. (2017). Automated hate speech detection and the problem of offensive language. In Eleventh international aaai conference on web and social media.
>
> [2]. Kerrigan, G., Smyth, P., and Steyvers, M. (2021). Combining human predictions with model probabilities via confusion matrices and calibration. Advances in Neural Information Processing Systems, 34.
>
> [3]. Hussein Mozannar, Hunter Lang, Dennis Wei, Prasanna Sattigeri, Subhro Das, and David A. Sontag. Who should predict? exact algorithms for learning to defer to humans . AISTATS 2023: 10520-10545

---

> > ### Comment · Reviewer_r3aW · 2023-08-16
> >
> > Thank you for the thorough response and providing additional experimental results. I do feel that my concerns have been well-addressed. Raised the score.

---

> > > ### Author Response · Authors · 2023-08-18
> > > **Thank you for increasing the score**
> > >
> > > We are glad to hear that we addressed your concerns. We are really appreciated for your valuable suggestion and time on our paper and thank you for updating the score!

---

### Official Review · Reviewer_fCzx · 2023-07-02

**Soundness:** 4 excellent
**Presentation:** 3 good
**Contribution:** 4 excellent
**Rating:** 7
**Confidence:** 4

**Summary:**

In this paper, the authors investigate the task of probability forecasting in multi-class classification with an expert deferral option (L2D). They address the issue of unbounded and invalid estimates of experts' accuracy that often arises in probability estimation for L2D. Furthermore, the authors highlight that this problem persists when using a loss function with a symmetric structure, such as the commonly used softmax cross-entropy loss. To overcome the problem of unbounded probability estimates and support the use of softmax parameterization in L2D, the authors introduce a novel asymmetric softmax parameterization. They demonstrate the consistency of the induced loss function and provide theoretical evidence that this loss function will never encounter the issue of unbounded estimation. The authors validate their proposed method through experiments conducted on multiple benchmark datasets, utilizing various evaluation metrics. The experimental results provide justification for the effectiveness of the proposed approach.

**Strengths:**

1.	The authors successfully enable the utilization of softmax parameterization in L2D, which is popular but discouraged in previous works. They provide both theoretical and experimental justifications for this approach. The proposed loss formulation is intuitive and straightforward to implement, requiring only the addition of a single output without modifying the scoring outputs of the classifier.

2.	The authors present a theoretical analysis highlighting the unbounded nature of symmetric losses in L2D. They demonstrate that the proposed loss function benefits from its asymmetric structure, emphasizing the non-trivial nature of their work.

3.	The authors offer theoretical guarantees regarding the proposed loss function, ensuring that it will never produce an unbounded probability estimator. This rigorous analysis eliminates any concerns regarding unbounded behavior in the proposed method.

4.	This work includes several figures that effectively compare the proposed method with related approaches. These figures illustrate the issues associated with unbounded probability estimators, providing persuasive evidence to aid readers in understanding the main points of the paper.

5.	The authors thoroughly validate the performance of the proposed method using a variety of evaluation metrics and settings. Their results demonstrate that the proposed approach is comparable to, if not superior to, baseline methods not only in the context of probability forecasting but also in predictive tasks without coverage constraints.


**Weaknesses:**

The authors are encouraged to provide a finite sample analysis, specifically an estimation error bound, to complement their theoretical analysis and ensure completeness. Considering that the newly proposed asymmetric softmax parameterization may alter the Lipschitz constant of the score function "g," the estimation error bound becomes a non-trivial aspect worth investigating. Additionally, combining this estimation error bound with the proposed regret transfer bound could potentially yield a more direct and intuitive result.

**Questions:**

In Theorem 2, it is proved that the proposed loss will never have an unbounded probability estimator. Though the proof is clear, can you provide a more intuitive explanation of such a positive result?

---

> ### Author Rebuttal · Authors · 2023-08-10
>
> **Q1. An estimation error bound should be given. The Lipschitzness of the asymmetric softmax w.r.t. g should also be clarified.**
>
> A1. Thank you for your helpful advice! We have derived the estimation error bound of the ERM with our proposed risk and will update it in the revised version of our manuscript. Though our proposed asymmetric softmax takes a different formulation, its Lipshitzness still persists: the absolute value of the partial derivative of $\tilde{\psi}_{i}(g)$ w.r.t.gj will not be larger than 1 for any i, j in [1,K+1], which indicates the Lipschitzness of the asymmetric softmax.
>
> **Q2. A more intuitive explanation should be given to Theorem 2.**
>
> A2. Thank you for raising this concern! A more intuitive explanation is that the existence of an unbounded probability estimator will remove some point g(x) from the potential solution. However, since any point can be mapped into the desired probability region with our proposed asymmetric softmax, any point my be the solution of point-wise risk w.r.t. some posterior probability, which leads to a contradiction. This explanation can also be seen as the scratch of the proof for Theorem 2.

---

> > ### Comment · Reviewer_fCzx · 2023-08-11
> > **To response**
> >
> > My concerns have been addressed. This is a good work! I am happy to accept this paper.

---

> > > ### Author Response · Authors · 2023-08-18
> > > **Thank you for supporting our work**
> > >
> > > Thank you for your constructive comments and suggestions.

---

### Official Review · Reviewer_hiuK · 2023-07-06

**Soundness:** 3 good
**Presentation:** 2 fair
**Contribution:** 3 good
**Rating:** 7
**Confidence:** 4

**Summary:**

This paper studies the learning to defer setup where you have to defer to an expert if the classifier is likely to be wrong. It has been shown that softmax based consistent estimators for the learning to defer losses do not provide calibrated probability estimates for the likelihood of deferring. One other work proposed non-softmax based losses which provided calibrated probability estimates but it is not known if any other softmax based loss can lead to calibrated estimates. This work shows that softmax is not the issue of the lack of calibration but using symmetric loss functions is. This work then proposes a softmax based asymmetric loss function that leads to calibrated and bounded probability estimates. This work also experimentally verifies the effectiveness of using their loss function.

**Strengths:**

- I find the idea of the paper really interesting. Since softmax based losses are the most commonly used loss functions, it makes sense to be able to use those loss functions for learning to defer framework. Moreover, this work identifies the fundamental issue present in the original loss function which was leading to poorly calibrated probability estimates. I find this insight on using asymmetric loss functions interesting.

**Weaknesses:**

Right now, the paper sometimes becomes hard to follow as there are a lot of notations used. One thing that could be improved is having a common notation section in the beginning which could be used to look up symbols. Overall, the presentation of the paper could be improved.

**Questions:**

In definition 1, R^c loss is not defined?

**Limitations:**

Yes

---

> ### Author Rebuttal · Authors · 2023-08-10
>
> **Q1. The presentation should be improved. It is encouraged to add a separated section to introduce the used notations.**
>
> A1. Thank you for your constructive advice! We will summarize the used notations in and add an extra section in the revised manuscript.
>
> **Q2. The R^c loss is not defined in Definition 1.**
>
> A2. Thank you for raising this concern! The $R^{c}_{01}$ should be the0-1-deferral risk instead and we will update it in the revised version of this manuscript.

---

> > ### Comment · Reviewer_hiuK · 2023-08-19
> > **Response**
> >
> > I thank the authors for addressing the concerns. I would like to keep my score.

---

> > > ### Author Response · Authors · 2023-08-20
> > > **Thank you for supporting our work**
> > >
> > > Thank you for your reply! We sincerely appreciate your valuable comments and time on our paper.

---

### Official Review · Reviewer_Jwha · 2023-07-06

**Soundness:** 3 good
**Presentation:** 3 good
**Contribution:** 2 fair
**Rating:** 6
**Confidence:** 4

**Summary:**

This paper shows that the miscalibration of the softmax-based surrogate loss for learning to defer is due to its symmetry. Instead, an asymmetric softmax-based surrogate loss is proposed and proved to be both calibrated and consistent. More generally, they reveal the connection between miscalibration and the symmetry of the used loss function and propose to design L2D surrogates based on asymmetric multi-class loss functions.

**Strengths:**

The paper is well-organized and enjoyable to read though the notation is a bit heavy. They not only give a careful analysis of softmax-based loss functions but also provide very interesting insights regarding the miscalibration of surrogate loss functions for learning to defer and the symmetry of the multi-class loss functions they are based on.

**Weaknesses:**

As stated in the paper, the 0-1 deferral loss studied here is not the most general version. For important settings including where the expert is a bigger model with higher accuracy but larger inference cost than the base model, an additional constant cost is required to reflect the extra inference cost. [31] shows that both [28] and [42] suffer from underfitting if the constant cost is non-zero. I am wondering if the proposed asymmetric softmax-based surrogate loss can be generalized to this setting and if it will have similar issues.

**Questions:**

1. Could you give more details on how Definition 3 is derived based on the asymmetric softmax function?

minors:
1. The notation of the risk is different in Definition 1 from in (1).
2. At the end of line 97, the transformation should map from R^{K + 1}.
3. It is better to specify the section when referring to the appendix.

**Limitations:**

See Weaknesses.

---

> ### Author Rebuttal · Authors · 2023-08-10
>
> **Q1. Can the proposed asymmetric softmax-based surrogate be generalized to the setting that an additional constant cost will be triggered when the model choose to defer to experts? If such generalization is available, will it have the same issue of underfitting as shown in [1]?**
>
> A1. Thank you for raising this concern! Since our surrogate can be seen as induced from a classification-calibrated multi-class classification surrogate, we can directly generalize it to the case where the additional cost $c_{0}$ is non-zero a based on the Proposition 2 of [1]:   $L_{\tilde{\psi}}(u,y,m)=-\sum_{i\in[K]}(\max_{j\in[K+1]}c(j)-c(i))\log(\tilde{\psi}_{i}(u))-(\max_{j\in[K+1]}c(j) -c(K+1))\log(\tilde{\psi}_{K+1}(u)) $.
>
> When the expert makes a wrong prediction, i.e., $\max_{j\in[K+1]}c(j)=1+c_{0}$, the surrogate can be further written as $L_{\tilde{\psi}}(u,y,m)=-\log(\tilde{\psi}_{y}(u))-c_{0}\sum_{i\in[K]}\log(\tilde{\psi}_{i}(u)) $. Notice that the second term is exactly a **label smoothing term**, which is potentially the root of underfitting as stated in [1] (in the second paragraph of Section 3.1). Based on this theoretical observation, we can infer that underfitting is likely to occur. I think combining the post-hoc method in [1] and designing surrogates that can be free from such label smoothing terms can be promising future directions.
>
> **Q2. The authors should give more details on how Definition 3 is derived based on the asymmetric softmax function.**
>
> A2. Thank you for raising this concern! To be detailed, we first map the scoring function from $R^{K+1}$ to $\Delta^{K}\cup[0,1]$ using the asymmetric softmax function. Then we can train a probability forecaster for classifier and expert accuracy simultaneously by minimizing the risk w.r.t. the sum of a multi-class classification surrogate and a binary classification surrogate, which is exactly Definition 3. The design of $\tilde{\psi}_{K+1}$ further binds the score of the deferral function and classifier, which lead to the property of maxima-preserving and ensure the consistency of Definition 3. We will elaborate on this point and add it to the paragraph next to Definition 3 in the future version.
>
> **Q3. There are some minor mistakes to be corrected and the section of appendix should be specified.**
>
> A3. Thank you for your constructive advice! We have revised our manuscript and will correct these clarity problems in the revised version of this manuscript.
>
> [1]. Mohammad-Amin Charusaie, Hussein Mozannar, David A. Sontag, and Samira Samadi. Sample efficient learning of predictors that complement humans. In ICML, 2022.

---

> > ### Comment · Reviewer_Jwha · 2023-08-17
> >
> > Thank you for the response. I have read the reviews and rebuttals. I agree that combining the proposed surrogate loss with the post-hoc method in [1] can be a promising future direction and will keep my score.

---

> > > ### Author Response · Authors · 2023-08-18
> > > **Thank you for keeping the positive score**
> > >
> > > Thank you for reading our response and keeping the positive score! We are really grateful for your time and expertise.

---

### Official Review · Reviewer_Ztxz · 2023-07-07

**Soundness:** 3 good
**Presentation:** 4 excellent
**Contribution:** 3 good
**Rating:** 7
**Confidence:** 3

**Summary:**

The paper studies the learning to defer (L2D framework), where one can defer to an expert decision when unsure about the model’s prediction, and a cost is incurred when either the prediction is wrong or when one defers to the expert and the expert makes a mistake.

The paper builds on top of prior work that shows that softmax parametrization can lead to unbounded probability estimators. While prior work provided a one-vs-all (OvA) formulation to solve this issue, this paper shows that the issue lies in the symmetric nature of the softmax function. The authors then proceed to introduce asymmetry in the softmax function and show that this leads to both bounded probability estimators and improved performance.


**Strengths:**

1. The paper is well-written and the idea is somewhat novel.
2. I enjoyed reading the theoretical analysis in both the main paper and the appendix.
3. I like how the method do not have a lot of additional hyper-parameters to tune — this makes the method simple and do not feel over-engineered.


**Weaknesses:**

1. Asymmetric softmax functions have studied before, although in a different context. For example, [1] suggests LDAM loss aimed for long-tailed or imbalance datasets. Comparison/discussion with this and possible other asymmetric variation of softmax would be appreciated.
2. The experiments involve only small-scale datasets like CIFAR-10 and CIFAR-100. While many methods perform well on these, they often do not scale well. Additional datasets would both (1) make the problem setting more appealing and (2) improve the standing of the paper.

[1] Kaidi Cao, Colin Wei, Adrien Gaidon, Nikos Arechiga, Tengyu Ma. Learning Imbalanced Datasets with Label-Distribution-Aware Margin Loss, NeurIPS 2019, https://arxiv.org/abs/1906.07413

**Questions:**

1. How is the learning to defer paradigm comparable to selective classification [1]? In particular, one can integrate the reject option into the classifier.
2. In the learning to defer problem, a possible generalization is also associating a cost (possibly small) with deferring the prediction, even when the expert is correct, because using an expert instead of the model should be costly and avoided. Curious if any related work has considered this scenario.


[1] Yonatan Geifman, Ran El-Yaniv. Selective Classification for Deep Neural Networks. NeurIPS 2017

---

> ### Author Rebuttal · Authors · 2023-08-10
>
> **Q1. The authors should add the comparison/discussion with [1] or any possible other asymmetric variation of softmax.**
>
> A1. Thank you for raising this concern! In this work, the asymmetric softmax is introduced to directly map the scoring function into the desired region $\Delta^{K}\cup[0,1]$ while remaining the order of each dimension of the scoring function. In [1], the softmax function is asymmetrized to simulate the trade-off between class margins that exist in imbalanced classification tasks, which can lead to smaller generalization errors if chosen properly. Though the motivations of the two versions of asymmetric softmax are different, they are both based on prior knowledge of the data distribution. In conclusion, introducing asymmetry into the original softmax function can be a promising solution when we are seeking methods to incorporate prior knowledge about data distribution into the learning process.
>
> **Q2. Additional datasets should be added to improve the standing and appealingness of this paper.**
>
> A2. Thank you for your valuable suggestion! To better validate the proposed method, especially on datasets with real-world expert information, we added additional experiments on HateSpeech and ImageNet-16H datasets, which are two datasets with real-world expert annotations, and the detailed information can be found in the one-page attachment.
>
> **Q3. The authors should compare the learning to defer paradigm with selective learning framework.**
>
> A3. Thank you for raising this concern! Learning to defer is a framework that focuses on point-wise rejection: any point whose maximum class-posterior probability is lower than the expert’s accuracy will be rejected. In contrast, the rejection rule in selective learning framework relies on the whole distribution with density $p(x,y)$: given a coverage constraint cov%, the rejection rule should accept the top-cov% samples with the highest maximum class-posterior probability. I think the goals of the two frameworks are not contradictory but complementary, and the combination of the two paradigms can be a promising future direction.
>
> **Q4. The authors are encouraged to give any related work that associates a cost with deferring the prediction.**
>
> A4. Thank you for raising this concern! Associating a cost $c_{0}$ with deferral has been studied in [2, 3], though some of the recent works [4, 5] on the surrogate losses for L2D only focus on the case where $c_{0}=0$. In this case, deferring to an expert will trigger a cost of $c_{0}+[[expert~prediction\not=y]]$. We find that our proposed method can be generalized to this case where $c_{0}>0$ and we provide the detailed formulation in the answer to question 2 of reviewer jwha.
>
> [1]. Kaidi Cao, Colin Wei, Adrien Gaidon, Nikos Arechiga, Tengyu Ma. Learning Imbalanced Datasets with Label-Distribution-Aware Margin Loss, NeurIPS 2019, https://arxiv.org/abs/1906.07413
>
> [2]. Hussein Mozannar and David A. Sontag. Consistent estimators for learning to defer to an expert. In ICML, 2020.
>
> [3]. Harikrishna Narasimhan, Wittawat Jitkrittum, Aditya Krishna Menon, Ankit Singh Rawat, and Sanjiv Kumar. Post-hoc estimators for learning to defer to an expert. In NeurIPS, 2022.
>
> [4]. Rajeev Verma and Eric T. Nalisnick. Calibrated learning to defer with one-vs-all classifiers. In ICML, 2022.
>
> [5]. Hussein Mozannar, Hunter Lang, Dennis Wei, Prasanna Sattigeri, Subhro Das, and David A. Sontag. Who should predict? exact algorithms for learning to defer to humans. AISTATS 2023: 10520-10545

---

> > ### Comment · Reviewer_Ztxz · 2023-08-11
> >
> > The authors have answered my questions and concerns in a compact manner, I thank them! To reflect this, I have increased the score from 6 to 7.
> >
> > Specifically:
> > 1. The authors have added experimental results on two more datasets: ImageNet-16H and Hatespeech.
> > 2. The authors have answered my question related to associating a cost with deferring the prediction.
> >
> > Additional suggestion:
> >
> > Similar to reviewer r3aW, I would suggest adding a few more datasets. For example, [1] uses synthetic expert data on CheXpert.
> >
> >
> > [1] Consistent Estimators for Learning to Defer to an Expert, https://arxiv.org/abs/2006.01862

---

> > > ### Author Response · Authors · 2023-08-18
> > > **Thank you for raising the score**
> > >
> > > Thanks for your recognition and suggestion. We are glad that you feel that the questions and concerns have been addressed. We will add the experiments on CheXpert and synthetic datasets in the final version.

---

### Author Rebuttal · Authors · 2023-08-10

## General Response

We thank all the reviewers for their valuable comments and devoted time. We are glad that all the reviewers praise the insight and theoretical contribution of this work. We are also encouraged that the reviewers find this work easy to use (Reviewers Ztxz, r3aW), and appreciate the clarity of this paper (Reviewers Ztxz, fCzx, jwha, and r3aW).

We respond to each reviewer's comments in details, respectively. An additional pdf file that includes one figure and three tables is provided to further experimentally validate our method and support our claims in the responses. In the revised version, we will update the manustript according to reviewers' suggestions. We believe this will definitely enhance the quality of this work.

---

### Decision · Program_Chairs · 2023-09-21

**Decision:**

Accept (poster)

**Comment:**

This paper addresses the open problem [42] of if there exists a well-calibrated softmax-based surrogate loss for learning-to-defer.  This paper demonstrates that the problem in the traditional softmax parameterization is due to symmetry, and a reasonable asymmetric version is proposed (that essentially applies a traditional softmax to the class dimensions and a map to [0, 1] to the defer dimension).  All reviewers have positive opinions of the paper, unanimously recommending acceptance.

Two good criticisms were put forward that the authors should take care to address in the camera-ready version.  The first is the need to have a softmax-based surrogate in the first place, as just because it is the default parameterization in deep learning does not mean it needs to be the default parameterization for multi-class learning-to-defer.  The authors provide a good rebuttal that the softmax can potentially provide probability estimates for all classes, whereas one-vs-all is unconstrained across classes.  I think this point and other motivations could be made more prominent in the draft (while simultaneously not leaving the reader to think the softmax will always dominate one-vs-all).  Secondly, Reviewer r3aW makes the good point that this paper's experiments are lacking compared to peer learning-to-defer papers.  The authors have promised to include experiments on HateSpeech and ImageNet-16H---as they should be sure to do.

My own question: can this loss be straightforwardly extended to the multi-expert setting, as the (symmetric) softmax seems to be an even bigger problem for calibration there [43]?  If so, then the authors should add the result at least to the appendix, as people might also be interested in the proposed fixed for the multi-expert setting.  If it is not obvious, then the authors should state this as an open problem / area for future work.